# Discovery of functional factorless internal ribosome entry site-like structures through virome mining

Subash Chapagain[1,2☉], Nicolas Salcedo-Porras[1,2☉], Amir Abdolahzadeh[1,2☉], Yaohua Zhang[1,2], Higor Sette Pereira[3], Stephane Flibotte[1,2,4], Kevin Low[1,2], Christina Young[1,2], Yuhang Wu[1,2], Shao Wang[1,2], Soh Ishiguro[5], Nozomu Yachie[5,6,7], Trushar Patel[3], Artem Babaian[8], Eric Jan [1,3]*

1 Department of Biochemistry and Molecular Biology, University of British Columbia, Vancouver, British Columbia, Canada, 2 Life Sciences Institute, University of British Columbia, Vancouver, British Columbia, Canada, 3 Department of Biological Sciences, University of Lethbridge, Alberta, Canada, 4 Bioinformatics Core Facility, University of British Columbia, Vancouver, British Columbia, Canada, 5 School of Biomedical Engineering, University of British Columbia, Vancouver, British Columbia, Canada, 6 Premium Research Institute for Human Metaverse Medicine (WPI-PRIMe), Osaka University, Suita, Osaka, Japan, 7 Research Center for Advanced Science and Technology, The University of Tokyo, Tokyo, Japan, 8 Department of Molecular Genetics, Donnelly Center for Cellular and Biomolecular Research, University of Toronto, Toronto, Ontario, Canada

☉ These authors contributed equally to this work.
* ej@mail.ubc.ca

## Abstract

All viruses must co-opt the host translational machinery for viral protein synthesis. The dicistrovirus intergenic region internal ribosome entry site (IGR-IRES) utilizes the most streamlined translation mechanism by adopting a triple pseudoknot structure that directly recruits and binds within the intersubunit space of the ribosome and initiates translation from a non-AUG codon. The origin of this unprecedented mechanism is not known. Using a bioinformatics pipeline to examine the diversity and function of IRESs across RNA viromes, we searched for IRES-like RNA structures using RNA covariance models for multiple IRES sub-types, and tested functional IRES by using a dual-fluorescent lentiviral library reporter screen. We identified over >4,700 dicistro-like genomes with ~32% containing putative IRES structures, including novel viral genome arrangements with multiple IRESs and IRESs embedded within open-reading frames (ORFs). Predicted IRESs bound directly to purified ribosomes and supported internal ribosome entry activity *in vitro* and *in vivo*. Moreover, internal IRESs embedded within an ORF of monocistronic genomes were functional and operated simultaneously to produce the downstream ORF. We also identified IRES-like structures within non-dicistrovirus viral genomes, including in the families *Tombusviridae* and *Narnaviridae* that bound to ribosomes directly and a subset can direct internal ribosome entry. This study provides a framework to map the origin of factorless IRES mechanisms and study the diverse viral strategies utilizing RNA-based mechanisms.

**Data availability statement:** All sequencing data are deposited in Jan, Eric; Chapagain, Subash (2025), "Discovery of functional factor-less internal ribosome entry site-like structures through virome mining", Mendeley Data, V1, doi: 10.17632/3rwdwcrfhw.1.

**Funding:** This study was supported by the UBC Life Sciences Institute Cores (Flow Flow Cytometry (ubcFLOW) and Bioinformatics Core facilities), which is supported by the UBC GREx Biological Resilience Initiative. This study was supported by a CIHR Project Grant (PJT-178342 to EJ) and BCIgnite grant (InnovateBC - IGNITE-2021-RND11-248-UBC-Jan-NanoVation to EJ), by the Canada Foundation for Innovation (CFI), the Canadian Institutes of Health Research (CIHR) CoVaRR-Net Project, and the Allen Distinguished Investigator Award to N.Y. N.Y. is supported by the Canada Research Chair program. T.R.P. is a Canada Research Chair in RNA and Protein Biophysics. H.S.P is supported by the RNA Innovation NSERC CREATE program (partially supported) and the Alberta Innovates Postdoctoral Fellowship. S.I. is supported by the Banting Postdoctoral Fellowship. We acknowledge infrastructure support from NSERC RTI and CFI grants. The SAXS data collection was supported by DIAMOND Light Source (SM36363), UK. The funders had no role in study design, data collection and analysis, decision to publish, or preparation of the manuscript.

**Competing interests:** The authors have declared that no competing interests exist.

## Author summary

The simplest eukaryotic protein synthesis mechanism to date is found within an internal ribosome entry site (IRES) from the dicistrovirus family. The IRES adopts an RNA structure that can directly bind to and manipulate the ribosome to initiate translation from a non-AUG codon. Using a bioinformatics pipeline, we identified >4700 viral genomes that have IRES-like structures, highlighting conserved elements and atypical genome arrangements that include an embedded IRES within viral open reading frames. Moreover, we identified IRES-like structure in other viral families suggesting widespread adoption of this viral translation strategy. This study highlights the diverse viral strategies utilizing RNA-driven mechanisms.

## Introduction

Viruses have evolved strategies to co-opt host translational machinery for viral protein synthesis [1]. Most eukaryotic mRNAs use a 5′ cap-dependent scanning translation initiation mechanism requiring at least 11 core translation initiation factors (eIFs) and 40S scanning to initiate translation from an AUG start codon. By contrast, some positive-strand RNA viruses use structured RNAs known as internal ribosome entry sites (IRESs) to promote 5′ cap-independent translation initiation [2,3]. IRESs are typically found within the 5′ UTR of some positive sense RNA viruses and in general, within the viral families *Picornaviridae*, *Caliciviridae* and *Dicistroviridae* [4,5]. IRES activities have also been reported in retroviruses HIV-1 and HIV-2, the thymidine kinase gene of herpes simplex virus, and in a few genomes of the family *Iflaviridae* [6–12]*,* thus, IRESs are widely-used across viral families.

In general, IRESs adopt an RNA structure that recruits a subset of translation initiation factors to drive viral protein synthesis that are classified based on sequence and structural homology, and the requirement of specific translation initiation factors to recruit the ribosome and initiate translation. To date, there are six viral IRES types: Type 1 and 2 IRESs exemplified by those in poliovirus and encephalomyocarditis (EMCV) genomes, respectively, require the majority of the core translation initiation factors and are stimulated by host IRES trans-acting factors such as polypyrimidine binding protein (PTB1) [13]. Type 1 and 2 IRESs are typically ~450 nucleotides in length and contain five structural domains. The main distinguishing property is that Type 1 IRESs involve 40S scanning to the AUG start codon whereas Type 2 IRESs recruit 40S subunits near the AUG start codon. Type 5 IRES, as exemplified by genomes of Aichiviruses, has similar factor requirements as Type 1 and 2 IRESs, however, adopts a distinct RNA structure that contains sub-domains of Type 1 and 2 IRESs and requires the host protein DHX29 for translation [14,15]. Type 3 IRESs such as the hepatitis A virus (HAV) IRES, direct translation requiring the cap-binding protein, eIF4E, and eIF4G, even though the HAV genome does not have a 5′ cap [16,17]. Type 4 IRESs such as the hepatitis C virus (HCV) and classic swine fever

virus (CSFV) IRESs, use a distinct translation strategy that can recruit the 40S subunit directly and uses only eIF3 and eIF2 to initiate translation [18]. The HCV IRES binds on the solvent side of the 40S subunit with distinct domains that interact with specific ribosomal proteins (e.g. uS7, eS27) and the ribosomal RNA expansion segment (e.g., ES7) and domain II protruding into the mRNA channel from the E-site of the ribosome [19]. Subclasses of Type 4 IRESs with minor differences in RNA structural domain compositions are found in the genomes of *Flaviviridae*, *Picornaviridae* and *Caliciviridae* [20,21].

The most striking IRESs are those found in Type 6 IRESs within genomes of dicistroviruses. Members of this family include the cricket paralysis virus (CrPV), honeybee-infecting Israeli acute paralysis virus (IAPV) and Taura syndrome virus (TSV), some of which have been associated with agricultural disease outbreaks [22,23]. Type 6 IRESs are located within the intergenic region of the positive-strand RNA genome between two main open reading frames (ORFs) that drive the translation of the viral structural proteins during infection (herein Type 6 IRESs may be referred to as intergenic IRES (IGR IRES)). Type 6 IRESs are ~180 nucleotides in length, adopting four stem-loops with overlapping three pseudoknots (PKI, PKII, PKIII) that recruit the ribosome directly (i.e., no eIFs required) and initiates translation from a non-AUG codon. Extensive biochemical and structural studies on model dicistrovirus IGR IRESs from CrPV, IAPV and TSV have led to mechanistic insights [24]. The IGR IRES uses two independent domains to recruit and drive translation: PKII and PKIII are responsible for binding to the 40S and 60S subunits and PKI mimics a tRNA anticodon-codon interaction [25]. The IRES binds within the intersubunit space of the ribosome occupying all three tRNA binding sites [26,27]. The conserved elements within the apical loops of SLV (CCGAC) and SLIV (AUUU) interact specifically with eS25 and eS5 to mediate 40S recruitment. The L1.1 bulge region mediates 60S subunit recruitment through interactions with the L1 stalk [28,29]. The PKI tRNA anticodon:codon mimicry domain initially occupies the A site of the ribosome [26]. The PKI domain undergoes translocation from the A to the ribosomal P site mediated by eEF2 thereby exposing the non-AUG codon in the A site for delivery of the first aminocyl-tRNA by the action of eEF1A, thus completing the first translocation step albeit uniquely without peptide bond formation. Subsequently, the PKI domain translocates from the P to E site of the ribosome mediated by another eEF2 [26,30–32]. At this second step of IRES-mediated translocation, the IRES undergoes a dynamic rearrangement within the E site of the ribosome whereby PKI anticodon-codon mimic flips from a tRNA anticodon stem to acceptor stem position [24]. In sum, the Type 6 IRES is an RNA translation factor that directly manipulates the ribosome into an elongation mode of translation.

The Type 6 IRESs can be further subdivided into six subtypes (6a-6f) that have distinct substructures and mechanisms of action (Fig 1). Type 6a and 6b IRESs are exemplified by the CrPV and IAPV IRESs, respectively. The main differences between the two IRESs are an extra stem-loop (SLIII) within the PKI domain of the IAPV IRES and conserved L1.1 sequences that are specific for each IRES subtype [28]. The SLIII of Type 6b IRESs mimics a tRNA acceptor stem within the ribosomal A site in IRES:ribosome complexes [27,32]. Despite these differences, both Type 6a and 6b IRESs use a similar mechanism to interact and drive translation. Indeed, the PKI domains of Type 6a and 6b IRESs can be swapped and still direct translation [33,34]. Type 6c-6F IRESs appear to use a distinct mechanism that can recruit pre-formed 80S ribosome, although they have distinct substructures. Type 6c-6f IRESs all contain a PKI anticodon-codon mimicry domain; however, each subtype has distinct PKII and PKIII domain [35–37]. Specifically, Type 6d-6f IRESs have a typical PKII base pair but the SLV domain is distinct and shorter than that in Type 6a and 6b and the PKIII base pair is within the apical stem of SLV. Type 6c, exemplified by the Halastavi arva virus IRES, is the most remarkable: this IRES lacks the SLV, SLIV and the PKIII domains [37]. Biochemical and cryo-EM structural studies showed that the HalV IRES binds to the 80S with the PKI domain occupying the ribosomal P site. Similarly, biochemical studies also showed that Type 6d-6f IRESs also start with the PKI in the P site, thus these IRESs occupy the P site to start translation from the A site [35,36]. In summary, the Type 6a-6f IRESs have revealed distinct ribosome assembly pathways and the diversity of factorless translation initiation mechanisms. Despite these mechanistic insights, the structural and functional differences within the Type 6 IRESs suggest diversion from a common ancestry. Further, the evolutionary origins of these Type 6 IRESs are not known.

**A**

Compile Dicistrovirus-like genomes from Serratus, NCBI, and metatranscriptomes (RdRP 50% AA ID)

↓

Genomes clustering at 98% nt ID

↓

Selection of genomes > 2kb that have 1 or more ORF > 1kb

↓

Prediction of Type 6 IRES subtypes (a, b, c, d, e, and f)

**B**

6a    6b    6c    6d    6e    6f

**C**

No. of predicted IRES structures

■ >6kb, with IRESs
□ <6kb, with IRESs

a: 1550 / 849
b: 543 / 206
c: 3 / 3
d: 624 / 452
e: 113 / 83
f: 128 / 98
ab: 168 / 113
ad: 127 / 107
df: 1 / 1

Subtypes

**E**

Genome count

□ Without IRESs
■ >6kb, with IRESs
□ <6kb, with IRESs

Genome size (kb): 2.0 4.0 6.0 8.0 10.0 12.0 14.0 >15.0

**D**

6a (849)
6b (206)
6c (3)
6d (452)
6e (83)
6f (98)
6a/6b (113)
6a/6d (107)
6d/6f (1)

N = 1912

Legend:
- 6a
- 6b
- 6c
- 6d
- 6e
- 6f
- 6a/6b
- 6a/6d
- 6d/6f

**F**

Triatovirus — HoCV-1, TrV, PSIV, BQCV, HiPV

Cripavirus Clade II — CrPV, DCV

Cripavirus Clade I — ALPV, RhPV

Aparavirus Clade I — ABPV, FEV1, KBV, IAPV, SINV-1

Aparavirus Clade II — TSV, MCV, MrTV

**Fig 1. Global discovery of Type 6 internal ribosome entry sites.** (A) Schematic of the bioinformatic pipeline to identify structures that are Type 6 IRES-like. Dicistrovirus genomes were selected from metatranscriptomes and the earth's virome project (Serratus.io). Transcripts containing an RdRP sequence with >50% amino acid identity from ICTV-classified dicistroviruses were selected using Palsmscan. Genomes were clustered at 98% nucleotide identity to eliminate redundant sequences, followed by selecting >2 kb genomes that contain at least 1 ORF > 1 kb in length (9,151 genomes, S1 Data). Type 6 RNA covariance models based on currently described IRES subtypes were generated and used to scan the final dataset using INFERNAL. (B) Secondary structure schematics of Type 6 IRESs. (C) Distribution of Type 6 IRES-like structures in dicistrovirus genomes by genome size. Total

genomes (light grey bars) and genomes containing IRESs (black bars) are shown. (D) Pie chart of the distribution of Type 6 IRES subtypes identified by INFERNAL. Genomes identified by multiple covariance models in the INFERNAL search are shown (S1 Data) (E) Distribution of Dicistrovirus genomes based on their genomic length. Shown in bar graphs is the number of genomes with or without IRES-like structures predicted by INFERNAL (F) Distribution of Type 6 IRES subtypes in an RdRP-based phylogenetic tree. RdRP amino acid sequences from genomes >6 kb were aligned using MUSCLE and a maximum likelihood tree was built with 1000 pseudo bootstrap branch support. IRES subtypes are identified with lines on the outside circle, colour-coded based on subtype. Sequences with multiple IRESs are shown with an extra stacked line. Genomes with IRESs that were used to build covariance models are shown on the outer ring. Virus names are specified for known dicistroviruses from the Cripavirus, Aparavirus, and Triatovirus genera (ICTV). Genomes and IRESs used to build the phylogenetic tree are listed in S1 and S2 Data.

Metagenomic studies in recent years have massively expanded the extent of the virosphere [38–40]. More recently, the mining of publicly available transcriptomes has further expanded the number of RNA viromes to greater than 130K RNA viromes [41]. The availability of these large metagenomic datasets provides an opportunity to explore the diversity and biological relevance of factorless IRES mechanisms within the family *Dicistroviridae*. In this study, we developed a bioinformatics pipeline to identify Type 6 IRES structures and validated the function of these predicted IRES-like structures. Moreover, these studies have revealed novel dicistrovirus genome architectures containing IRES structures, thus providing insights into novel viral strategies that use the IRES to drive viral protein synthesis. Finally, we provide evidence that the factorless IRES mechanism is widespread beyond the family *Dicistroviridae*.

## Results

### Dicistrovirus type 6 IRES diversity

To explore the diversity of Type 6 IRES in dicistroviruses, we compiled dicistrovirus genomes from the Earth's Virome Project (Serratus.io), NCBI-annotated virus genomes, and metagenomic studies [38,41]. We used the RNA-dependent RNA polymerase (RdRP) protein sequences from known dicistroviruses (International Committee on Taxonomy of Viruses:ICTV) as a taxonomic barcode to scan these datasets in initially selecting viromes with an RdRP identity of at least >50% at the amino acid level compared to known dicistrovirus RdRP using Palmscan (Fig 1A). In total, we identified 64,245 transcripts containing dicistrovirus-like RdRP sequences. To eliminate redundant and short sequences, we clustered sequences at a 98% nucleotide identity and excluded sequences below 6,000 bases in length and with at least one ORF > 1000 nucleotides (i.e., > 333 codons), thus maximizing the identity of full-length dicistrovirus genomes, considering that dicistrovirus genomes are ~ 9,000 nt in length. This curation resulted in a final list of 4,704 dicistrovirus-like genomes used for subsequent analyses (S1 Data).

To identify genomes with Type 6 IRES-like structures (Fig 1B), covariance models of the six known Type 6 IRES subtypes were generated and used to scan the dicistrovirus genome database using the covariance model algorithm INFERNAL [42] (Fig 1A). A total of 3257 Type 6 IRES-like structures were identified (~33% of genomes), among these, 1912 IRESs were unique in their nucleotide sequence, while 1345 IRESs were found in multiple genomes (Fig 1C and S1 Data). Moreover, the bioinformatics pipeline identified previously-described subtype 6a and 6b IRESs including IGR IRESs from the genera Cripavirus (CrPV, Drosophila C virus (DCV), Nilaparvata lugens C virus (NLCV), Rhopalosiphum padi virus (RhPV), and aphid lethal paralysis virus (ALPV)), Triatovirus (Triatoma virus (TrV), Himetobi P virus (HiPV), Plautia stali intestine virus (PSIV), and black queen cell virus (BQCV), and Aparavirus (acute bee paralysis virus (ABPV), Kashmir bee virus (KBV), IAPV, Formica exsecta virus 1 (FEV1), TSV, Macrobrachium rosenbergii Taihu virus (MrTV), mud crab virus (MCV)) thus providing further confidence and validating this bioinformatics approach (Fig 1D and S1 Data). Of the 1912 unique IRES-like structures identified, the majority were IRESs of subtypes 6a (849, 44.4%), 6b (206, 10.7%) and 6d (452, 23.6%), whereas a smaller fraction of subtypes 6c (3, 0.1%), 6e (83, 4.3%), and 6f (98, 5.1%) IRES structures were identified. Interestingly, a fraction of the identified IRES structures was recognized by more than one covariance model subtype, 113 (5.9%) IRES structures were classified as both subtype 6a and 6b IRESs, 107 (5.6%) as both subtype 6a and 6d IRES structures, and 1 (0.05%) as both subtype 6d and 6f IRES structures (Fig 1C and 1D). In general, these

mixed subtype IRES-like structures have low INFERNAL scores, containing conserved elements such as the SLIV, SLV, and PKI and deviate in other regions.

While the bioinformatics pipeline identified IRES-like structures in a third of the genomes, IRES-like structures were not identified in the other genomes. The absence of IRES-like RNA structures in the majority of dicistrovirus-like genomes could be attributed to incomplete or truncated genomes. Most detected IRESs were present in genomes larger than 6 kb (Fig 1E and S1 Data). In this subset, 2051/4294 (50.1%) of the genomes contained candidate IRES-like structures. This fraction remained constant in genomes >7,000 nucleotides in length (1948/3754, 51.9%), or >8,000 nucleotides in length (1662/3140, 52.9%). Similarly, the majority of dicistrovirus genomes that do not contain an Type 6 IRES-like structure within the intergenic region were >6 kb in length.

To gain insights into the evolution of Type 6 IRESs, we constructed an RdRP-based phylogenetic tree of the dicistrovirus genomes (<6 kb) and plotted the predicted IRES-like structure subtypes in this tree (Fig 1F and S1 Data). Genomes with subtype 6a IRES structures were distributed in multiple clades throughout the tree, suggesting this IRES structure may be closer to the ancestral Type 6 IRES and/or that distant viral genomes obtained IRESs via recombination events. Genomes from viruses of the Cripavirus genus, that have been described to contain 6a IRESs, were separated into a basal clade (containing RhPV and ALPV and a derived clade including DCV, NLCV and CrPV), confirming that this is paraphyletic [43]. In contrast, the Triatovirus genus was in a single monophyletic group. Genomes with subtype 6b IRES structures were mostly clustered in two clades; a large basal clade (containing Solenopsis invicta virus-1 (SINV-1), ABPV, KBV, FEV1, and IAPV) and a smaller derived clade (containing TSV, MrTV, and MCV). This distribution confirmed that this is likely a paraphyletic group [43]. Type 6c IRES structures were the rarest, present in only two closely related genomes. Most genomes with a subtype 6d IRES structure clustered in a monophyletic clade. Genomes with subtype 6e were closely related in a monophyletic clade thus suggesting that this IRES subtype may be present only in specialized dicistrovirus genomes. Genomes with subtype 6f IRES structures were in a monophyletic clade, clustered within the subtype 6d IRES structure clade, hinting that the 6f subtype may have derived from subtype 6d IRESs. IRES-like structures that were identified by multiple covariance models (6a-6b, 6a-6d, and 6d-6f) did not show clustering into monophyletic or nearly monophyletic clades, suggesting that these may be recently evolved IRES structures that have not been positively selected or that the chimeric IRES structures occurred via recombination.

## IRES consensus models

To examine conserved elements within the predicted Type 6 IRES-like structures, we aligned unique high-scoring IRES subtypes through multiple sequence alignments to construct consensus models (Fig 2, top 50% INFERNAL scores of each subtype, S2 Data). Type 6a IRES structures, as exemplified by the CrPV IRES, (Fig 2) showed high conservation in the internal loop 1.1 (L1.1), stem-loop IV (SLIV), and the stem-loop V (SLV), which was not unexpected as these domains are essential for the Type 6a IRESs to interact with ribosomal subunits; the L1.1 interacts with the L1 stalk from the 60S subunit [26,29,44], the SLIV interact with the rps7 and the SLV with the rps25 from the 40S subunit [26,27,30]. Conservation was also observed within the PKI domain; VLR, PKI base pairing and L3.1 mediate specific steps of IRES translation including frame selection and translocation [45,46]. The Type 6a IRES consensus alignment further identified conserved regions within adenines in the L1.2 close to the PKII base pairing, UA-rich single-stranded regions (S2.2 and S2.3) upstream of the SLV, residues in the helical P1.1 region, and a CG pairing within PKII. From structural analysis of the CrPV IRES and IRES:ribosome complexes, the adenines within L1.2 are involved in A-minor interactions with the minor groove of the P2.2 helix [26,29].

Type 6b IRESs (Fig 2), as exemplified by the IAPV and TSV IRESs, adopt a structure similar to 6a IRESs, containing three PKs and uniquely an additional stem-loop close to the PKI (SLIII). The predicted Type 6b IRES structural model showed high levels of conservation in SLIV and SLV, which have similar interactions with the 40S ribosomal subunits as in Type 6a IRES:40S complexes [27]. Surprisingly, within the L1.1 region of 6b IRES model, only a few nucleotides in the

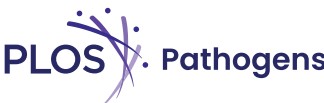

**Fig 2. Consensus models of Type 6 IRES.** Secondary structure models of indicated Type 6 IRESs are shown with colour-coded heat map of nucleotides representing percent conservation from alignment of the top 50% scoring sequences for each subtype based on INFERNAL output (S2 Data). The

range of base pairing are also denoted and the number of sequences used for alignment of each sub-type are indicated. Pie charts show the percentages of the predicted codon adjacent to the pseudoknot I (PKI) IRES sequences. The resulting multiple sequence alignments were manually curated to remove gaps.

distal part of the internal loop were strongly conserved, despite the role of this region for ribosome assembly on the IRES [28,47]. Nucleotides surrounding the SLIV showed moderate levels of conservation (S2.2, S2.3, and P2.2) whereas L1.2, L3.2, VLR, the linker S2.5, and the junction of L3.1, L3.2, and L3.3 showed a higher level of conservation. The L3.1, L.3.2 and L3.3 junction is important for +1 frame translation in a subset of 6b IRESs [48]. L3.3 and SLIII showed low levels of conservation despite their reported role in promoting IRES-mediated translation.

For predicted Subtype 6d IRES structures, conservation was observed within SLIV and L1.1, domains that interact with both 40S and 60S subunits (Fig 2). Other regions included two adenosines within L1.2a. For predicted subtypes 6e and 6f IRES structures, conservation was observed in L1.1 and P2.2, and within PKI. Subtype 6f IRES-like structures showed some conservation within the three nucleotides within S2.4. Subtypes 6d-6f IRESs have been shown to recruit 80S ribosomes directly, similar to subtype 6c IRESs [35–37]. In general, these analyses highlighted the length of the base-paired regions and key nucleotide identity conservation that may be important for IRES function.

A unique feature of IGR IRES mechanism is that the IRES initiates translation from a non-AUG codon adjacent to the PKI domain. We examined the putative start codon usage of the predicted Type 6 IRES-like structures (Fig 2 and S2 Data). For subtypes 6a to 6d, the dominant predicted start codon was an alanine codon with GCU being the most prevalent start codon. There was a strong preference for a G at the first nucleotide of the starting codon. Structural studies of IGR IRES:ribosome complexes have shown that this G base adopts intrastrand stacking interactions within the PKI loop, thus contributing to the anticodon-codon mimicry of PKI [26,32]. For predicted subtype 6f IRES structures, the most prevalent putative start codons were ACA, CCU and ACU, from highest to lowest (Fig 2). These analyses suggested a strong selection of putative start codons for IGR IRES-mediated viral protein synthesis.

## Functional validation of predicted IRES structures

We next examined whether the bioinformatically-predicted IGR IRES structures are functional. To address this, we cloned a library of 1111 of the 1912 predicted Type 6 IRESs within the intergenic region of a lentivirus dual fluorescent protein reporter containing an upstream eGFP and downstream mRuby3 (Fig 3A and S3 Data). eGFP and mRuby3 are driven by cap-dependent and IRES-mediated translation mechanisms, respectively. The cloned sequences were selected by clustering the 1912 Type 6 IRESs at 90% nucleotide identity to prevent overrepresentation of highly similar sequences (S3 Data). Lentivirus-transduced HEK293T cells were then treated with the ER stress inducer, thapsigargin, which is a known treatment known to increase dicistrovirus IGR IRES translation [49,50]. Cells were subjected to flow cytometry analysis (FACS) and cell-sorted for both eGFP and mRuby3 positive cells. Under basal conditions, approximately ~1.3% of the cells displayed both high eGFP and mRuby3 fluorescence suggesting that relatively few IRESs were active (Fig 3B). By contrast, in Thapsigargin-treated cells, the number of cells displaying both high eGFP and mRuby3 increased to ~4.1% of cells, which is consistent with increased IRES translation under ER stress [50]. Cells displaying high and low mRuby3 (i.e. IRESs that were active in both basal and Thapsigargin-treated) were isolated and genomic DNA was purified, from which the IRES region was PCR amplified and identified by next-generation sequencing. A total of 209 IRES-like structures were identified, the majority of which were enriched of subtype 6a and 6b IRESs (Fig 3C and S3 Data). In general, there was an enrichment of IRES structures (i.e., more read counts) in thapsigargin-treated cells over untreated cells (Figs 3C and S1C and S3 Data). Plotting the fold-change (mRuby3 positive/mRuby3 negative) enrichment of IRES structures in untreated (x-axis) vs thapsigargin-treated cells (y-axis) revealed a negative correlation (S1A Fig), indicating that IRESs that were not translated under basal conditions were relatively more actively translated under ER stress.

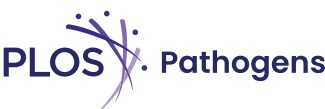

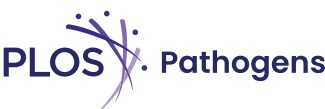

**Fig 3. Functional analysis of IRES-like RNAs.** (A) Schematic of IRES screening pipeline in mammalian cells. A library of IRESs were cloned within the intergenic region of a lentivirus eGFP-mRuby3 dual reporter construct. HEK293 cells were transduced with lentivirus and fluorescent cells were analyzed



by FACS analysis and sorted. Cells were either untreated or treated with Thapsigargin (1 µM) for 18 hours. (B) FACS plots of cells eGFP (y-axis) and mRuby3 (x-axis) fluorescence. The percent of mRuby3 positive cells are shown under specific cell treatments. IRESs identified in untreated and Thapsigargin treated cells are listed in S3 Data. (C) Sequencing read counts for IRESs in mRuby3 positive cells transduced with lentiviral reporter library in untreated (red) and Thapsigargin treated (blue) conditions. The counts are ranked from highest to lowest for IRESs enriched in untreated conditions. All raw read counts for unfiltered and filtered IRESs are in S3 Data. (D) *In vitro* translation. Select Type 6a and 6b IRESs were tested in (left graph) rabbit reticulocyte lysates (1 hour, 37°C) or (right graph) in Sf21 insect lysates (2 hours, 30°C) using *in vitro* transcribed dual luciferase reporter RNAs. IRESs selected are listed in S4 Data. Shown are averages ± s.d. from at least three independent experiments of FLuc/RLuc luciferase activity ratios normalized to that of bicistronic RNAs containing the wild-type (WT) CrPV IGR IRES. mPKI represents mutant CrPV IRES containing mutations that disrupt PKI base pairing. INFERNAL scores are shown below the indicated IRESs. Shown are averages ± s.d. from at least three independent experiments.

To further confirm that the predicted IRESs are functional, we selected candidate Type 6a and 6b IRES structures from the FACS-sorted fluorescent cell screen and the bioinformatics pipeline (i.e., high INFERNAL scores) (S1, S3 and S4 Data). We chose a range of subtype IRES-like structures and cloned them within the intergenic region of the dual luciferase reporter constructs (Fig 3D and S4 Data). For simplicity, we re-labeled the IRES subtypes 6a-1 to 6a-11 and 6b-1 to 6b-9 (S4 Data). The IRES sequences, the genomes from which the IRESs were identified, the INFERNAL predicted score and the corresponding predicted secondary structure models are shown in S4 Data and S2 Fig. *In vitro* transcribed RNAs were incubated in rabbit reticulocyte lysate (RRL) or insect Sf21 extracts and luciferase activities were measured. As expected, the reporter RNA containing the wild-type CrPV IGR IRES led to Firefly luciferase activity (shown as Firefly:Renilla (FLuc/RLuc) ratio) indicative of active IRES translation, whereas a mutant CrPV IRES containing mutations that disrupted PKI base-pairing did not (Fig 3D). Normalizing the translational activities to that of the wild-type CrPV IRES, the select Type 6a and 6b IRESs displayed a range of translational activities (Fig 3D). Approximately half of the IRES structures (6a-14 and 6b-15) showed IRES activities equal to or ~two-fold higher as compared to the CrPV IRES in RRL (Fig 3D). Although the majority tested showed IRES translation in both RRL and Sf21 extracts, the level of FLuc/RLuc ratios were not as high in Sf21 extract as those observed in RRL. Furthermore, some IRESs displayed opposite translational activities in RRL versus Sf21 extracts (Fig 3D; see 6a-1, 6b-1, 6b-2 IRESs as examples), suggesting that there may be species-specific differences that govern IRES translation *in vitro*. These results conclusively validated the bioinformatics approach in identifying *bona fide* dicistrovirus IRES elements.

To determine whether the predicted RNA structures are important for IRES function, we generated mutations that disrupt PKI base-pairing within select active Type 6a and 6b IRESs and tested them using the dual luciferase reporter assay *in vitro* in RRL (Fig 4A). As predicted, disruption of the predicted PKI base-pairing abolished IRES translation whereas compensatory mutations that restored PKI base-pairing partially rescued IRES activity (Fig 4A and 4B - representative 6b-2 IRES model; S2 Fig for models of each IRES), similar to that observed with other Type 6 IRESs [51,52]. These results demonstrated that the integrity of the predicted PKI structure is important for IRES translation.

We next examined whether the IRESs can bind to ribosomes directly, which is a key unique property of Type 6 IRESs [49]. Select [$^{32}$P]-end labeled Type 6a and 6b IRES RNAs were incubated with increasing concentrations of purified salt-washed 40S and 60S subunits. Of note, we chose IRESs that were active and inactive in *in vitro* translation extracts (Fig 3D) and were of relatively high INFERNAL scores. All IRESs tested bound directly to 80S ribosomes with an apparent $K_D$ ranging from 2.5-12 nM (Fig 4C and 4E), which is within the range of known IGR IRES:ribosome binding affinities [25,28]. We examined the Type 6a-11 IRES structure in more detail and investigated if mutations that disrupt PKII and/or PKIII base pairing reduced IRES binding to the ribosome (Fig 4C, left bottom). As expected, PKII and PKIII mutant versions of this IRES impaired binding to the 80S ribosome, indicating that the integrity of the predicted IRES structure directed ribosome binding. To examine if the IRES structure is important for translation in a more biologically relevant context, we tested the IRES activity in insect S2 cells by transfection of 5′ cap bicistronic reporter RNA (Fig 4D). The wild-type IRES supported translation in S2 cells, whereas the PKI mutant (mPKI) showed significantly reduced translational activity (Fig 4D). In summary, we have demonstrated that the bioinformatics pipeline approach identified *bona fide* IRES-like structures that can assemble ribosomes directly and initiate IRES translation.

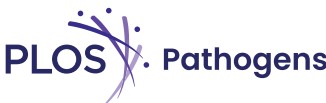

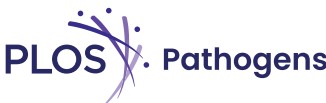

Fig 4. Integrity of IRES structure for translation and ribosome binding. (A) Integrity of PKI for IRES function. Bicistronic luciferase reporter RNAs containing select IRESes containing either mutations that disrupt PKI base pairing (mPKI) or compensatory mutations that restore PKI base pairing (cPKI)

were incubated in RRL for 1 hour, 37°C. Shown are averages of FLuc/RLuc luciferase activity ratios normalized to that of bicistronic RNAs containing the wild-type (WT) CrPV IGR IRES. mPKI represents mutant CrPV IRES containing mutations that disrupt PKI base pairing. (B) Secondary structure model of the Type 6b-2 IGR IRES. Mutations that disrupt the PKI base pairing and compensatory mutations that restore the base pairing are indicated in rectangular boxes. (C) 80S ribosome binding. Filter binding assay of purified, salt washed human 40S and 60S incubated with select [$^{32}$P]-labeled RNAs (0.5 nM) and the fraction of IRES:ribosome complexes were measured by phosphorimager analysis. mPKII/PKIII and mPKI/mPKII/mPKIII represent mutant IGR IRESs whereby PKI, PKII and PKIII base pairing are disrupted. (D) Apparent dissociation constant ($K_D$) values of 80S:IRES complexes. (E) Relative luciferase activity of Type 6a IRES subtype in S2 cells. S2 cells were transfected with reporter RNAs containing the indicated wild-type 6a-11 IRES or CrPV IRES (WT), mPKI (mutation in IRES that disrupts PKI base pairing). Shown are averages of Firefly:Renilla (Fluc:RLuc) luciferase activities at 6 hours post-transfection. All data are averages±s.d. from at least three independent experiments.

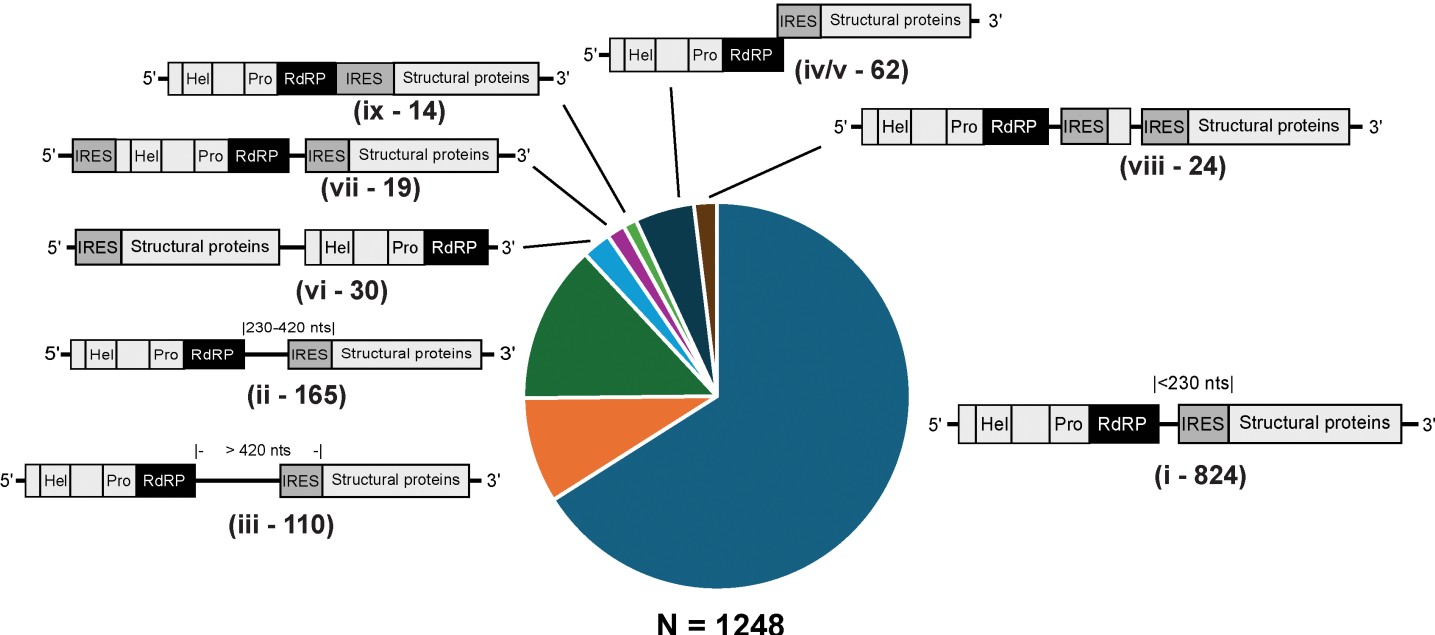

**Fig 5. Genome architectures of dicistrovirus-like genomes.** Genome organization of dicistrovirus genomes was obtained by predicting ORFs using ORffinder, RdRps with Palmscan, and IRESs with INFERNAL. ORF2 > 1 kb in length were predicted using any possible non-stop codon as a start codon. The number of genomes within specific sub-types and the total numbers are shown to the right. Genomes are categorized with predicted IRESs located within the 5′ UTR or a single long ORF, or genomes with multiple predicted IRESs. Genomes containing predicted IRES with more than two ORFs are shown. Predicated helicase (Hel) and 3C-like protease (Pro) and RNA-dependent RNA polymerase (RdRP) are shown. Genomes are listed in S5 Data.

## Novel dicistrovirus genomic architectures

We next analyzed the location of the predicted IRES structures within the viral genome. We examined viral genomes that were >6 kb in length in order to capture near complete genomes (S1 and S5 Data). Of the 805 genomes that contained predicted unique IRES structures, the majority of genomes (532/805) resembled architectures of known dicistroviruses harboring a single IRES within the intergenic region (Figs 5 and S3). Of these genomes, the IRES structures can be grouped based on the location within the intergenic region: (i) the majority of IRES structures (824) are adjacent to or close to the ORF1 stop codon (intergenic region is < 230 nucleotides including the predicted IRES structure), whereas 275 genomes ((ii) and (iii)) contained IRES structures downstream of the ORF1 stop codon that are within an intergenic region of 230–420 nucleotides or >420 nucleotides (Figs 5 and S3). These subgroups of IRES-containing IGRs can be further divided into specific IRES sub-types: most are of Type 6a and 6b IRESs (S3 Fig).

Interestingly, a fraction of genomes ((iv), (61 genomes) had the stop codon of ORF1 within the predicted IRES structure (Figs 5 and S3). This genomic arrangement is reminiscent of the honeybee dicistrovirus genomes, such as IAPV and ABPV, that contain a stop codon within a 5' stem-loop that promotes IRES translation [51,53]. Within this subgroup, subtype 6a and 6d IRES structures were predominant. Unexpectedly, there was one genome which has an extended ORF1 whereby the stop codon is downstream of the IRES structure and putative start site (v) (Figs 5 and S3).

Unlike IRES-bicistronic genome architectures described above, we also identified other atypical IRES genome arrangements (Figs 5 and S3). We identified 30 genomes (vi) that contained a predicted Type 6 IRES structure within the 5' UTR, of which they were similar to Type 6a and 6b subtype IRESs. Moreover, we also identified a small subset of genomes (19 genomes, (vii)) with predicted Type 6 IGR IRES structures within both the 5' UTR and the intergenic region. Most of these genomes contained subtype 6a IRES structures in both the 5' UTR and IGR, whereas a few genomes contained mixtures of subtype IRES-like structures (S3 Fig). Remarkably, we identified 24 genomes containing two predicted IRES structures within the intergenic region (viii). In these cases, the upstream IRES structure is predicted to drive translation of a small ORF ranging between 93–286 amino acids in length, whereas the downstream IRES structure directs translation of a downstream ORF. Finally, and surprisingly, there was a small subset of genomes that were monocistronic and contained a predicted embedded IRES structure overlapping within the coding region (14 genomes, (ix)). In these genomes, the organization and order of non-structural and structural proteins were similar to other dicistrovirus genomes [54]. The putative embedded IRES is downstream of the RdRP and is predicted to drive translation of the viral structural proteins.

## Functional characterization of IRESs within atypical dicistrovirus genomes

The location of the predicted IRES structures in atypical dicistrovirus genome architectures suggests that the IRESs may be utilized in distinct ways to regulate the stoichiometry of viral protein expression during infection. We chose two distinct IRES/genome architectures for follow-up experiments: the genome with the ORF1 stop codon downstream of the predicted IRES structure (v) and genomes that are monocistronic and contain an overlapping IRES (ix). The location of the IRES structure embedded within the viral open reading frame poses the question of how translation of ORF1 or the monocistronic main ORF affects IRES-mediated translation.

The Type 6a-2 IRES is functional (Fig 3D) and is located within the genome whereby the stop codon (UGA) of ORF1 is directly downstream and adjacent to the putative PKI domain (Fig 6A). The RdRP domain is located upstream of the IRES whereas the translated protein sequence encoded within the IRES did not yield any obvious known structure or sequence via Blast or Alphafold3. We hypothesized that ribosomes that translate ORF1 unwind the IRES and thus, inactivate IRES-mediated translation. To address this, we generated three dual bicistronic reporters in which the 6a-2 IRES is cloned between RLuc and FLuc luciferase. In the first reporter (Fig 6B, WT), the 6a-2 IRES is situated to mimic the natural genome arrangement whereby the RLuc ORF extends into and past the IRES and the stop codon is located downstream of the IRES structure. Here, scanning-dependent translation of the RLuc and IRES-driven FLuc would express ~47 kDa and ~65 kDa proteins, respectively (Fig 6B). The second reporter (RS; RLuc stop codon) is similar to the first reporter but contains a stop codon before the IRES, thus generating a bicistronic reporter. The third reporter contains a strong hairpin structure (HP) within the 5' UTR and thus blocks scanning-dependent translation. We tested these reporters in *in vitro* translation extracts monitoring expression of RLuc and FLuc by incorporation of [35S]-methionine. As shown previously, bicistronic RNAs containing the wild-type but not mutant PKI CrPV IGR IRES, resulted in expression of FLuc (Fig 6C). Bicistronic RNAs containing the 6a-2 IRES in its natural context resulted in translation of both the ~47 kDa and ~65 kDa proteins. Similarly, FLuc expression was also detected in bicistronic RNAs containing a stop codon downstream of RLuc. Insertion of a strong hairpin within the 5' UTR abolished scanning-dependent RLuc translation as expected but resulted in higher IRES-mediated FLuc translation (~1.5 fold) compared to the other two WT and RS bicistronic reporters (Fig 6C and S4 Data). These results indicated that the IRES is functional though it is located within the ORF.



**Fig 6. Functional analysis of IRES within an atypical dicistrovirus genome.** (A) Schematic of the Type 6a-2 IRES (type 6a-2) within the atypical dicistrovirus genome. Predicted secondary structure model of the Type 6a-2 IRES. (B) Schematic of reporter RNAs mirroring their locations in atypical dicistrovirus genome. "RS" denotes a stop codon at the end of the Renilla luciferase ORF (stop codon is represented by a red octagon). "HP" denotes a strong hairpin stem-loop inserted within the 5′ UTR. The predicted translated protein masses are shown above. (C) SDS-PAGE analysis of *in vitro* translation reactions (top) and quantification of proteins (bottom). *In vitro*-transcribed bicistronic reporter RNA (600 ng) containing the indicated wild-type or mutant IRESs were incubated in RRL (1 hour, 37°C) with [35S]-methionine. (D) IRES-mediated translation in mock and CrPV-infected (MOI 10) S2 cells. S2 cells were transfected with reporter RNAs containing the indicated wild-type 6a-2 IRES or CrPV IRES (WT), mPKI (mutation in IRES that disrupts PKI base pairing), RS (Renilla with a stop codon) or HP (hairpin within the 5′ UTR). Shown are averages of Firefly (FLuc) and Renilla (RLuc) luciferase activities ± s.d. from at least three independent experiments at 6 hours post-transfection.

To validate these results, we monitored translation in S2 cells by transfecting *in vitro* transcribed capped-bicistronic mRNAs. In these experiments, we infected S2 cells with the dicistrovirus, CrPV, to simulate virus infection which activates IRES activity. Cells transfected with the bicistronic RNA containing the stop codon after RLuc (RS) resulted in higher IRES-mediated translation as compared to that of the reporter RNA containing the 6a-2 IRES in its natural context (WT) (Fig 6D). These results are consistent with the model that ribosomes may translate through the overlapping IRES and reduce IRES translation, possibly by unwinding and disrupting IRES structure. However, these results cannot rule out that there is competition between cap- and IRES-dependent translation within the bicistronic reporter.

### Embedded IRES-like structure in the viral ORF

An unexpected finding from the bioinformatics search was the discovery of embedded Type 6 IRES structures within the viral monocistronic open frame (Fig 5ix and S5 Data). These overlapping IRES structures (herein called contiguous or

embedded IRES) are located downstream of the RdRP and are predicted to drive translation of viral structural proteins in ORF2 (Fig 7). There were sixteen genomes containing embedded IRES structures, which were all of either subtype 6a or 6b IRESs (Figs 5 and S2). We cloned these IRES structures into the dual luciferase reporter construct whereby the RLuc and FLuc were fused in frame with the contiguous IRES structure (Fig 7B, schematic of bicistronic reporters). Thus, translation of the full-length RLuc-IRES-FLuc would produce a 106 kDa protein (Fig 7B). Reporter RNAs with contiguous IRES structures led to translation of the FLuc 64 kDa protein *in vitro* (Fig 7C) with varying degrees of expression (ranging from 5 to 100% as compared to that of the wild-type CrPV IRES), thus demonstrating that most of these contiguous IRES structures are functional within the context of an ORF. Interestingly, scanning-dependent translation of the full-length polyprotein (RLuc-IRES-FLuc, 106 kDa protein) was also observed for the majority of contiguous IRES-containing reporter RNAs *in vitro*, suggesting that both scanning-dependent and IRES-mediated translation can occur (Fig 7C). Moreover, a reporter RNA with contiguous IRES 6a-9, resulted in expression of a truncated RLuc (36 kDa protein) which may suggest that ribosomes may be dropping off upon reaching the IRES structure (Fig 7C).

Using the subtype 6a-11 embedded IRES structure as a model (Fig 7A), we used several approaches to further confirm that this embedded IRES structure is functional. First, introducing a stop codon directly upstream of the IRES (and downstream of the RLuc) still led to translation of FLuc (Figs 3C and 7D, compare lanes 1 and 2 (Ren-Stop)). Second, mutations that disrupt PKI base pairing (mPKI), all three PK base pairing (mPKI-III) or mutations within SLIV, SLV and L1.1 (mSL4) all abolished FLuc expression (Fig 7D, lanes 3–6).

Third, to conclusively confirm IRES translation, we tested whether the contiguous 6a-11 IRES can function in a circular RNA (circRNA). Because the contiguous IRES is devoid of in-frame stop codons in the native polyprotein ORF, we reasoned that the IRES could drive continuous translation (i.e., a circRNA without a stop codon) [55]. Towards this, we generated an IRES-NLuc circRNA that does not contain an in-frame stop codon, and thus ribosomes would continuously drive translation of multiple NLuc proteins (WT-NLuc-circ) (Fig 7E, schematic of circRNA reporters and expected protein products). Further, we generated reporters whereby two porcine teschovirus 2A (P2A) "stop-go" sequences flank NLuc in order to release individual NLuc proteins (WT-P2A-NLuc-circ). To monitor translation, we tested RNAs in RRL with [35S]-methionine to follow protein synthesis. In RRL, the WT-NLuc-circ RNA led to the expression of a smear of high molecular weight proteins (>80 kDa), consistent with rolling circle translation (Fig 7F, lane 3). In contrast, the insertion of flanking P2As of NLuc or addition of a stop codon downstream of NLuc (and thus, preventing circRNA translation) resulted in the expression of ~20 kDa NLuc (Fig 7F, lane 4 and 5). To confirm that rolling circle translation was occurring, we generated circRNA that contains flanking suboptimal P2A sequences that lack the N-terminal GSG sequence of P2A (WT-subP2A-NLuc-circ) [56]. This circular RNA WT-subP2A-NLuc-circ resulted in two predominant bands at ~20kDa and ~40 kDa, which is consistent with inefficient 2A peptide activity during rolling circle translation (Fig 7F, lane 2). Mutations that disrupt PKI (mPKI) or all three PK (3mPK) base-pairing inhibited translation, albeit there was still minor expression in reactions containing the mPKI mutant, suggesting that an intact PKII and PKIII can still recruit the ribosome and direct circRNA translation *in vitro* (Fig 7F, lanes 5 and 6). In summary, we showed that the 6a-11 IRES can direct rolling circle translation.

We next examined rolling circle translation by the contiguous 6a-11 IRES in transfected S2 cells. Transfection of IRES-containing circular RNAs resulted in higher NLuc activity as compared to the linear versions (Fig 7G). Of note, the circular RNAs that permitted rolling circle translation displayed higher NLuc than that of the WT-NLuc circRNA which contains a stop codon (WT-Nluc-stop-circ) (Fig 7G). The circRNA displaying the highest NLuc activity was the WT-P2A-Nluc-circ, followed by the WT-NLuc-circ and WT-sub2A-NLuc-circ. These results indicated that protein processing by the 2A peptide affects the NLuc translation and enzymatic activity. The mutant IRES containing disruptions of all three PKs, abolished translation (Fig 7G, 3mPK). By contrast, mutations that disrupt PKI base pairing (mPKI), which inhibits ribosome positioning on the IRES but not ribosome binding [25], still displayed NLuc activity (Fig 7G), suggesting that ribosome recruitment by the IRES with intact PKII and PKIII basepairing can still direct rolling circle translation. In summary, the contiguous IRES can direct internal ribosome entry translation initiation in the context of a reporter circRNA.

**A** Contiguous IRES within dicistro-like ORF

**B** Bicistronic Reporter RNA

**C**

**D**

**E**

**F**

Lanes
1- No RNA
2- WT-subP2A-NLuc-circ
3- WT-NLuc-circ
4- WT-P2A-NLuc-circ
5- WT-NLuc-stop-circ
6- mPKI-subP2A-NLuc-circ
7- 3mPK-NLuc-circ
8- mPKI-NLuc-circ

**G** S2 cells

**Fig 7. Translation of embedded contiguous IRESs.** (A) Genomic location (top schematic) of a representative embedded IRES (type 6a-11) and its predicted secondary structure (bottom). P2A – denotes 2A peptide from porcine tescovirus-1. Scar – denotes sequence left over after RNA

circularization. (B) Schematic of the luciferase reporter RNAs containing the IRES embedded within the ORF mirroring the location of the IRES in the native monocistronic genome. Predicted MW of RLuc (36.0 kDa), FLuc (61.7 kDa) and fusion RLuc-IRES-FLuc (106 kDa) are shown. Stop codon is indicated as a red octagon. (C) Translation of luciferase reporters containing select embedded IRESs (listed in S4 Data). *In vitro* transcribed bicistronic reporter RNAs containing the indicated IRESs were incubated in RRL (1 hour, 37°C) with [$^{35}$S]-methionine. Reactions were loaded on 12% SDS-PAGE gels and imaged by phosphorimager analysis. Representative gels (top panel) and quantification of relative Firefly luciferase band intensities (bottom graphs) are shown. (D) Mutational analysis of Type 6a-11 IRES. Representative SDS-PAGE gel of [$^{35}$S]-methionine *in vitro* translation reactions of reporter RNAs in RRL. mPKI-III denotes mutations that disrupt PKI, PKII and PKIII base pairing, m4SL denotes mutations within SLIV, SLV and L1.1 collectively. "Stop ORF1" denotes a luciferase reporter wherein a stop codon is inserted at the end of Renilla luciferase as shown in (B). (E) Schematic of circular RNAs and predicted circRNA protein products. (F) Representative SDS-PAGE gel of *in vitro* translation (RRL) reactions containing [$^{35}$S]-methionine and the indicated circRNA reporter. "SubP2A" denotes P2A sequence lacking the N-terminal GSG. 3mPK denotes mutations that disrupts all three pseudoknots within the IRES. (G) Translational activity of IRES circRNA in S2 cells. S2 cells were transfected with circular RNAs containing the embedded IRES. At 8 hours post-transfection, cells were harvested and lysed and luciferase activities were measured. Shown are averages ± s.d. from at least three independent experiments.

### Factorless IRESs beyond family *Dicistroviridae*

Given that the bioinformatics pipeline identified *bona fide* factorless IRESs, we next asked whether IRES-like structures can be identified in other RNA viral genomes. Searching the RNA virome from the NCBI database and Serratus using INFERNAL, we identified eight IRES-like elements in positive-sense single-stranded genomes of *Picornaviridae*, *Marnaviridae*, *Narnaviridae* and *Tombusviridae*, and of these, six were Type 6a and two were Type 6b IRESs (S2 Fig for models and S4 Data). Of note, the predicted start sites of these IRESs were in-frame of the downstream viral ORF, thus strongly suggesting that these putative IRESs direct translation in these non-dicistrovirus genomes. For instance, in the picornavirus and marnavirus genomes, the IRES-like structures are located within the intergenic region between two main ORFs, presumably to direct the translation of the downstream structural proteins (Fig 8A). In the narnavirus genome, the IRES is located within the 5′ UTR upstream of the single polyprotein ORF (Fig 8A). The IRES-like structures in the tombusvirus genomes were all located within the intergenic region upstream of the ORF encoding the capsid (Fig 8A). During the replication cycle of tombusvirus, a subgenomic RNA containing the capsid ORF is generated, which is typically translated by a 3′ UTR translational element called a CITE (cap-independent translation enhancer) [57]. Given the location of the IRES-like structures within the tombuvirus-like genomes, they would also be predicted to drive the translation of the tombusvirus capsid ORF.

Using the bicistronic reporters, we tested whether these non-dicistrovirus IRES-like structures can drive translation in RRL and insect Sf21 extracts (non-DV-1 to -8) (Fig 8B). Four of the eight IRESs directed FLuc expression and notably two of them, one from picornavirus and one from narnavirus, displayed higher IRES activity than that of the CrPV IGR IRES (Fig 8B). Despite the non-dicistrovirus IRES-like structures having relatively high INFERNAL scores, it was intriguing that the four putative IRESs from tombusvirus-like genomes did not support IRES activity (non-DV-1–4 IRES-like RNAs). We reasoned that given that tombusvirus typically infects plants, IRES-like RNAs may be functional in a plant extract or that the RNAs may be directing translation in alternative reading frames as a subset of dicistrovirus IRESs can drive alternative reading frame translation [51]. However, the non-DV-1 and non-DV-2 RNAs from tombusvirus-like genomes did not support translation in wheat germ extracts, nor could direct translation in all three frames (S4 Fig).

We next addressed whether the non-DV IRES-like RNA adopts an RNA structure as predicted by INFERNAL. To follow-up, we performed SAXS and SHAPE-seq analysis of the non-DV-1 IRES-like RNA (S5 Fig and S6 Data, see details of analysis). Briefly, fifty models of the non-DV-1 RNA were calculated with a X2 value of approximately 1.07, indicating a good fit of the experimentally collected scattering data with the *ab initio* envelopes. These models were then averaged and filtered to derive a single representative model for each RNA using DAMAVER, which exhibits an NSD value close to 1 (S5E and S5F Fig), implying that the average model closely resembles all generated models. We used the SHAPE reactivity profiling to gain insights into the secondary structure profile of non-DV-1 RNA. The SHAPE reactivities of each nucleotide were mapped and represented as a folded structure in S5G Fig. Overall, nucleotides throughout the RNA





**Fig 8. Functional IGR IRES in non-dicistrovirus viral genomes.** (A) Schematic of non-dicistrovirus genomes and the location of predicted Type 6 IRES. (B) Non-dicistrovirus-like IRES translation. *In vitro* transcribed dual luciferase reporter RNAs containing the indicated non-dicistrovirus predicted

IRESs within the intergenic region were incubated in RRL (1 hour, 37°C) or Sf21 extracts (2 hours, 30°C). Luciferase activities were measured and the ratio of FLuc/RLuc was calculated and normalized to that of CrPV IRES containing reporter RNAs. (C) Genome organization of Tombusvirus-like non-DV1 genome with a predicted IGR IRES (top) and secondary structure model of the putative IRES (bottom). Mutations used for subsequent binding experiments are denoted in the boxes. (D) Predicted non-DV1 RNA structure model from SAXS data (light blue) overlapped with atomistic structure, represented in ribbons. (E) Sucrose gradient centrifugation analysis (top) showing percent total radioactive counts (y-axis) in fractions (top to bottom) of reactions containing [$^{32}$P] labeled IRES incubated with purified salt-washed 40S and 60S. Wild-type (black lines) and mutant (grey lines) of CrPV IRES and non-DV-1 RNA are shown. Mutant IRES denotes mutations that disrupt both PKII and PKIII base pairing. Filter binding assays (bottom) of purified human 80S with indicated wild-type and mutant IRES. $^{32}$P labeled IRESs (0.5 nM) were incubated with increasing concentrations of purified, salt-washed human 40S and 60S subunits and the fraction of IRES:ribosome complexes were measured by phosphorimager analysis. Shown are averages ± s.d. from at least three independent experiments.

structure showed high reactivities (> 0.4). The secondary structure constraint provided by SHAPE, combined with the three-dimensional SAXS envelope, provided insights into the high-resolution structural organization of non-DV-1 RNA. Fig 8D shows the fitting of the SAXS envelope (surface structure in light blue) with the SHAPE-derived computational structure (dark cyan). These structures were overlapped using DAMSUP and presented an NSD of 1.07 (Figs 8D and S5E), indicating a good fit. Additionally, we used the "fit in map" function of ChimeraX to calculate the correlation coefficient between these two structures, which is 0.7749. This statistical parameter confirms the good fit, indicating a strong and positive correlation between the structures. Based on these structural analyses, the non-DV-1 RNA adopts an RNA structure that is reminiscent of a Type 6a IRES, as predicted by the INFERNAL pipeline.

Given that non-DV-1 adopts a Type 6a IRES structure, we next asked whether the non-DV-1 RNA binds directly to ribosomes. Using sucrose gradient centrifugation, we fractionated reactions containing [$^{32}$-P]-labeled RNA and purified salt-washed ribosomes. As shown previously, the wild-type but not mutant (TM, triple PKI/PKII/PKIII mutation) CrPV IRES was detected in the 80S fraction (Fig 8E, fractions 23–24), indicating direct ribosome binding the IRES. Similarly, the non-DV-1 IRES-like RNA was present in the 80S fraction. Mutating PKII and PKIII base pairing of non-DV-1 IRES-like RNAs abolished 80S binding (Fig 8E). To further validate these findings, we monitored 80S binding to the non-DV-1 RNA by filter-binding analysis. The non-DV-1 RNA bound to purified 80S ribosomes with an apparent $K_D$ 23.45 ± 4.16 nM whereas mutating PKII and PKIII base-pairing impaired 80S binding (Fig 8E). These results indicated that the non-DV-1 IRES-like RNA structure can mediate direct ribosome binding. Overall, these data revealed IGR-IRES like RNA elements in viral genomes other than dicistroviruses and that IRES-driven mechanism may be more widespread in virosphere.

## Discussion

The Type 6 IRESs of dicistroviruses use the simplest eukaryotic translation initiation mechanism. From extensive molecular, biochemical and structural studies into this factorless IRES mechanism, it is now apparent that there are IRES subtypes that mediate distinct ribosome assembly pathways: subtypes 6c-6f IRESs recruit 80S ribosomes directly and can direct translation from the ribosomal P site [35–37], thus hinting that there may be other IRES mechanisms to be identified. In this study, we developed a bioinformatics pipeline to identify Type 6 IRES-like structures and demonstrated that this pipeline identifies *bona fide* functional IRESs that can mediate factorless ribosome recruitment and translation. This pipeline has expanded the number of factorless IRES structures and provides a phylogenetic framework to delve into the origins of this mechanism. Furthermore, the bioinformatics pipeline identified functional Type 6 IRESs in non-dicistrovirus genomes, thus suggesting that this IRES mechanism is used more widely beyond a single viral family as a strategy to drive viral protein synthesis.

INFERNAL identified functional Type 6 IRES-like structures and provides a framework for improving RNA structure predictions. However, the correlation between INFERNAL scores and IRES activity was only moderately correlated (S1A and S1B Fig). Comparing the active versus inactive IRESs showed that the inactive IRESs tend to adopt structures containing unpaired nucleotides or bulges within double-stranded RNA domains (S2 Fig). It is possible that some IRESs may be more active in more physiologically relevant systems such as the host cells that support the specific virus infections. In

addition to using INFERNAL which primarily uses a molecular co-variation prediction algorithm, applying thermodynamic and hierarchical RNA folding considerations for RNA secondary structure and pseudoknot predictions along with biochemical structural probing approaches should lead to more refined IRES RNA structural predictions [42,58–60]. Furthermore, co-variation phylogenetic analysis can further highlight specific functional substructures that have evolved within the IRES [61]. Finally, our bioinformatics approach did not identify Type 6 IRES-like structures in the majority of dicistrovirus genomes, which may suggest novel RNA structures and/or translation mechanisms. Indeed, our previous study showed that a whitefly dicistrovirus genome utilizes a factor-dependent intergenic IRES mechanism [62].

Type 6a and 6b IRESs were the most prevalent Type 6 IRESs found in the family *Dicistroviridae*. Moreover, the Type 6a and 6b IRESs were identified within specific dicistrovirus clades (Fig 1), thus suggesting that these IRESs may have evolved from a common ancestral IRES structure. However, Type 6a and 6b IRESs were also present scattered in other clades, suggesting recombination events. This observation is in line with previous reports that the IRESs are modular and that some domains can be functionally exchanged [33,37,63]. The general consensus sequence models of Type 6a and 6b IRESs highlight the relevance of distinct domains that support biochemical and structural studies (Fig 2, examples in L1.1, SLV and SLIV and the VLR). By contrast, Type 6d-6f IRESs also displayed consensus sequences and domains that were distinct from those from Type 6a and 6b, which reflects the distinct mechanisms. Further structural studies of these subtype IRESs bound to the ribosome will provide insights. Of note, Type 6d-6f IRESs were found in distinct dicistrovirus clades suggesting distinct evolutionary trajectories.

An interesting finding was the identification of atypical architectures of dicistrovirus genomes and the location of the IRESs. Most of the genomes were typical of known dicistrovirus genomes with the Type 6 IRESs contained entirely within the intergenic region (Fig 5 and S5 Data), thus highlighting that the two main ORFs are regulated independently: IRES drives translation of the downstream structural ORF independently of the upstream ORF [54,64]. This viral strategy allows independent control of the two ORFs at the temporal level and stoichiometric expression of structural versus nonstructural proteins. However, the location of the IRESs in atypical dicistrovirus genomes were enlightening and suggests that the location of the IRESs is used for distinct viral protein synthesis strategies. For example, the Type 6 IRESs were identified within the 5′ UTR only or within both the 5′ UTR and the intergenic region (Fig 5vi and 5vii). These genomes could still utilize the IRES mechanisms independently to control the expression of the ORF1 nonstructural and ORF2 structural proteins. In extreme cases, two IRES were found within the intergenic region whereby the upstream Type 6 IRES drives translation of a small viral ORF (Fig 5viii), thus suggesting that two IRESs can function within the intergenic region of the two main ORFs and drive translation of the small ORF to increase coding capacity. There was a subset of IRESs whereby the upstream ORF1 stop codon is located within the IRES or downstream of the IRES (Fig 5iv and 5v). These architectures suggest the possibility of translational coupling whereby the translation of ORF1 may affect IRES-driven translation. Our results may support this model as inhibiting ORF1 translation led to an increase in IRES-mediated translation of ORF2 (Fig 6C), however, more studies will be needed to formally rule out the alternative model whereby cap- and IRES-dependent translation are competing within the bicistronic reporter. Further studies will need to be investigated to determine the impact of having a stop codon in these contexts, and whether there is a benefit as a translational coupling strategy or a means of unfolding the IRES for the purpose of stoichiometric balance. Finally, the identification of functional IRESs embedded within the viral polyprotein was striking (Fig 7). Although there have been limited reports describing embedded IRESs within host [65] and viral ORFs [66], this work is the first to demonstrate a functional embedded IRES within a dicistrovirus viral ORF. An outstanding question is whether translation of the main ORF and the embedded IRES translation can occur from the same mRNA transcript. Our studies showed that both cap-dependent and IRES translation products can be detected *in vitro* (Fig 7), which begs the question of how translation of the main ORF and IRES-mediated translation may occur at the same time. Mutations in the IRES that disrupt the secondary RNA structure or ribosome binding abolished IRES activity but did not affect translation of the main ORF *in vitro*, which suggests that unwinding activity of the ribosome and/or from cellular helicases may be sufficient for the ribosome to translate through the IRES (Fig 7D).

Further investigation using single molecule mRNA translation studies may provide insights into the mechanistic relationship of full-length and embedded IRES translation [67]. In summary, these studies have provided hints of diverse viral strategies using the IRES to control viral protein synthesis in many ways.

IRES types (ex. Type 1, 2, 4 and 5) have been detected widely in distinct viral families, such as in members of *Picornaviridae*, *Flaviridae* and *Caliciviridae*, indicating that horizontal exchange of IRESs can occur between genomes of viral species [20,21]. Further, the exchange of functional modular RNA domains through recombination events can also lead to new IRES classes with chimeric IRES types, a prediction that has been experimentally tested for Type 4 and Type 6 IRESs [33,37,58,68–70]. The identification of Type 6 IRESs beyond the family *Dicistroviridae* further support that horizontal transfer via recombination events have been acquired by other viral families as a strategy to drive viral translation. Moreover, on top of validating that a subset of these non-dicistrovirus IRESs are functional (Fig 8), the location of the IRESs within these genomes strongly suggests that these IRESs may be important for viral protein synthesis. However, there were exceptions. Even though the Type 6 IRES-like RNAs from tombusvirus-like genomes were identified and can bind to ribosomes directly, they did not support translation *in vitro* in RRL and wheat germ extracts (Figs 8 and S3). Tombusviruses, which infect plants, typically use a 3′ UTR translation mechanism called a 3′ CITE to recruit translation factors to promote translation. A recent study has shown that the 3′ CITE can direct cap-independent translation [71]. It is possible that the Type 6 IRES, which is located in the 5′ UTR of the subgenomic RNA may act in synergy with a 3′ CITE. However, it is noted that the 3′ UTRs of these tombusvirus-like genomes were not fully complete or assembled, as the 3′ UTRs were much shorter than typical [72]. It is also possible that these Type 6 IRESs function only in the specific host species that these viruses infect or may have an undetermined function for virus infection.

The recent discovery and expansion of the RNA virosphere has provided a treasure trove of novel viruses and genome architectures, thus setting the stage to identify novel viral RNA structures and strategies [38–40]. Indeed, recent reports have uncovered a multitude of novel viruses, including viroid-like circular RNAs and 3′ UTR elements that contribute to viral RNA stability [41,73–76]. By concentrating on specific RNA structures and translation mechanisms in this study, we have identified novel IRESs and genome features, thereby establishing a foundation for uncovering new viral translation mechanisms and evolutionary strategies.

## Materials and methods

### Plasmids

Dual and monocistronic luciferase reporter constructs were generated as described [77]. Briefly, IRESs were synthesized (Twist Biosciences) and cloned within the intergenic region of bicistronic construct or within the 5′ UTR of monocistronic construct. The library of predicted IRESs were synthesized (Twist Biosciences) and cloned into the intergenic region of a lentivirus reporter expression plasmid containing eGFP and mRuby3.

### Tissue culture cells

HEK293T cells transduced with lentivirus were maintained in DMEM supplemented with 10% fetal bovine serum (FBS) at 37°C. Drosophila Schneider S2 cell line (S2) cells were maintained in Shields and Sang M2 insect medium (Sigma-Aldrich, St. Louis, MI, USA) supplemented with 10% (FBS) at 25°C.

### *In vitro* transcription

Bicistronic and monocistronic DNA were linearized with BamHI and NcoI (NEB) restriction enzymes respectively, and RNAs were *in vitro* transcribed in reactions with T7 RNA polymerase as described [77]. Integrity of the RNA was monitored by gel analysis and quantitated by Nanodrop spectrophotometer. *In vitro* transcribed RNA was 5′ capped post-transcriptionally (Cellscript).

For SAXS analysis, the plasmid constructs were purified and then digested using XbaI (NEB, Canada). *In vitro* transcription (IVT) reactions were carried out under the control of the T7 RNA polymerase promoter, and the synthesized RNA was subsequently purified using size exclusion chromatography (SEC), following previously published protocols [63,78]. The purified fractions were eluted using an RNA buffer (10 mM Bis-tris pH 6.0, 100 mM NaCl, 15 mM KCl, 5 mM MgCl2, and 5% glycerol) and concentrated through overnight ethanol precipitation. The resulting pellets were then resuspended in RNA buffer to achieve a concentration of approximately 1.5 mg/mL for downstream applications. The integrity of the RNA was assessed using nondenaturing agarose gel electrophoresis.

### *In vitro* translation assays

*In vitro* transcribed RNAs were incubated in rabbit reticulocyte lysates (RRL) or Sf21 insect lysates (Promega) for 1 hour at 37°C or 30°C in the presence of [$^{35}$S]-methionine as described previously [78]. Reactions were either resolved on 12% SDS-PAGE gel and analyzed by phosphorimager analysis (Typhoon, Amersham, Chicago, IL, USA) or analyzed for enzymatic luciferase activity (Promega) using a dual luciferase kit (Promega) in Tecan Spark-10M Multimode plate reader.

### Transfection

S2 cells ($2.5 \times 10^6$ cells) were transfected with bicistronic RNA (2 µg) using Lipofectamine 2000 reagent (Thermo Fisher Scientific, Waltham, MA, USA). For monitoring reporter RNA translation under CrPV infection, S2 cells were transfected with bicistronic RNA for 1 h, followed by infection with CrPV (MOI = 10) for 4.5 h. Cells were harvested at the indicated time and lysed in 1X passive lysis buffer (Promega). Lysates were cleared and protein concentration was measured by Bradford assay (Bio-Rad, Hercules, CA, USA). Equal amounts of total protein were used to measure luciferase enzymatic activity according to the manufacturer's protocol (Promega).

### Purification of the 40S and 60S subunits

HeLa cell pellets (Cell Culture Company) were used to purify ribosomes as described [25]. Briefly, HeLa cells were lysed in a lysis buffer (15mM Tris-HCL (pH 7.5), 300mM NaCl, 6mM MgCl2, 1% (v/v) Triton X-100, 1mg/mL heparin). Debris was removed by centrifuging at 23,000~g and the supernatant was layered with 30% (w/w) cushion of sucrose in 0.5M KCl and centrifuged at 100,000~g to pellet crude ribosomes. Pelleted ribosomes were gently resuspended in Buffer B (20mM Tris-HCL (pH 7.5), 6mM magnesium acetate, 150mM KCL, 6.8% (w/v) sucrose, 1mM DTT) at 4°C. Next, the ribosomes were treated with puromycin (final concentration 2.3mM) to release mRNA and KCl (final concentration 500mM) was added to wash and dissociate 80S ribosomes into 40S and 60S. The dissociated ribosomes were then separated on a 10%-30% (w/w) sucrose gradient. Absorbance at 260 nm was measured to detect the 40S and 60S peaks and corresponding fractions were pooled and concentrated using Amicon Ultra spin concentrators (Millipore Sigma) in buffer C (20mM Tris-HCl (pH 7.5), 0.2 mM ED buy TA, 10mM KCL, 1mM MgCl2, 6.8% sucrose). The conversions 1 A260 nm = 50 nM for 40S subunits, and 1 A260 nm = 25 nM for 60S subunits were used to determine the concentration of 40S and 60S subunits by spectrophotometry.

### Assembly and analysis of ribosomal complexes by sucrose density gradient centrifugation

10 pmol [$^{32}$P]-labeled IGR IRES mRNA were incubated with 30 pmol 40S subunits and 40 pmol 60S subunits in buffer E (final concentration: 20 mM Tris pH 7.5, 100 mM KCl, 2.5 mM MgOAc, 0.25 mM Spermidine and 2 mM DTT) for 20 min at 30°C to form ribosomal complexes, and then resolved by centrifugation through 10%-30% sucrose density gradients in buffer E in Beckman SW55 rotor at 55,000 rpm for 90 min at 4°C. Optical density of fractionated gradients was measured at 260nm, and the presence of radioactivity was monitored by Cherenkov counting in a scintillation counter (Beckman LS 6000IC).

## Ribosome filter binding assays

Ribosome:IRES complexes were monitored by filter binding assays as described [33]. Briefly, *in vitro* transcribed RNAs (final 0.5 nM) were preheated at 65 °C for 3 min and 1X buffer E was added and slow cooled in a preheated water bath for 20 min. Pre-folded RNAs, along with non-competitor RNA were incubated with an increasing concentration of 40S and 60s subunits at room temperature for 15 minutes. Reactions were then loaded onto a Bio-Dot filtration apparatus (Bio-Rad) with a double membrane of nitrocellulose and nylon pre-washed with buffer E. The membranes were then washed thrice with buffer E, dried, and the radioactivity was imaged and quantified by phosphorimager (Amersham) analysis. The dissociation constant was calculated by the formula

$$[AB]/[A]total = fmax[B]/([B] + KD)$$

where [A] is the concentration of RNAs, [B] is the concentration of ribosomes, [AB] is the concentration of RNAs bound to the ribosomes, fmax is the saturation point, and $K_D$ is the dissociation constant.

## SHAPE RNA structural probing

For SHAPE-MaP analysis of the minimal IRES, *in vitro* transcribed RNA for non-DV1 IRES was used as described [79]. Briefly, 500 ng of RNA was heated to 95° for 3 min, followed by the addition of Buffer E (final concentration of 20-mM Tris, pH 7.5, 0.1-M KCl, pH 7.0, 2.5-mM MgOAc, 0.25-mM spermidine and 2-mM dithiothreitol (DTT) and incubated at 30°C for 20 min. Folded RNA was modified by adding N-methylisatoic anhydride (NMIA) dissolved in DMSO (final concentration 5mM) and incubated for 45 minutes at 30°C. Control reactions containing only DMSO (no NMIA), as well as an additional denaturing control (DC) reactions were performed in parallel. The RNA was recovered by ethanol precipitation. RT Primer extension of modified and control RNAs was performed with primer 5'-GTGTTGTTGTTTTGTTGTTTTCGTTG-3' using Superscript III Reverse Transcriptase (SSIII, Thermo Fisher Scientific), with Mn2 +; followed by second strand cDNA synthesis (NEB #E6111). The cDNA was then used for subsequent library preparation and sequencing. Libraries were sequenced on an Illumina MiSeq platform following the manufacturer's standard cluster generation and sequencing protocols. SHAPE-Mapper 2.0 was used to analyze the sequencing data as described [80]. RNAstructure [81] was used to predict and model the secondary structure using the SHAPE-MaP data.

## Small-angle X-ray scattering (SAXS)

SAXS data was collected at the B21 beamline in Diamond Light Source, located in Didcot, Oxfordshire, UK. The collection was carried out using an Agilent 1200 high-performance liquid chromatography (HPLC) system with a specialized flow cell, as previously outlined [82]. A Shodex KW403-4F SEC column was employed at a flow rate of 0.160 mL/min with RNA buffer. X-ray exposure lasted for 3 seconds over 600 frames. Analysis of the data was conducted using the ATSAS suite [83]. CHROMIXS [84] was initially utilized on each raw data set to account for buffer contribution. Guinier analysis was employed to assess the radius of gyration (Rg) and the homogeneity of the samples (q2 vs. ln(I(q))) [85]. The folding state of the RNA was determined using dimensionless Kratky analysis (qRg vs. qRg2*I(q)/I(0)) [86]. Paired distance distribution (P(r)) plots were generated to establish the real space Rg and the maximum dimension (Dmax) of the RNAs [87]. DAMMIN [88] was used to produce twenty models for non-DV-1 RNA based on the P(r) data. Subsequently, DAMAVER and DAMFILT [89] were employed to generate an average and filtered model for each RNA using these models. SHAPE-Mapper 2.0 was used to analyze the sequencing data as described [80]. RNAstructure [81] was used to predict and model the secondary structure using the SHAPE-MaP data.

## Atomistic RNA structure determination

Reactivity data from SHAPE-MaP was employed to determine the secondary structure. The SHAPE sequencing map's reactivity profile was analyzed using RNAstructure, and a dot-bracket secondary structure profile was visualized through StructureEditor [81]. The lowest free energy structure from the SHAPE analysis was chosen and used as secondary structure constraints in SimRNA v3.20 [90] to generate three-dimensional tertiary models for each RNA. A total of 20000 models for non-DV-1 RNA were produced. DAMSUP was utilized to align the tertiary models of each RNA with the SAXS models and calculate the normalized spatial discrepancy (NSD). Finally, the UCSF ChimeraX-1.8 [91] fit-in map tool was employed to determine the correlation coefficient between computational models and the SAXS envelope. The best representative models were filtered and chosen based on NSD and correlation coefficient values and visual inspection of structural fitting. The models were visually depicted using ChimeraX.

## IRES lentiviral screen

The IRES library cloned into the lentiviral dual reporter was transformed into competent bacteria until a 10X coverage of colonies to inserts was achieved. Colonies were pooled and plasmids extracted for lentiviral infection. Cells were subjected to FACS under basal conditions and after 18h of exposure to 1 μM of Thapsigargin, selecting both eGFP and mRuby3 positive cells.

GFP and mRuby3 (putative IRES activity) and GFP-only positive cells (putative inactive IRES) were subjected to DNA extraction. Inserts were PCR amplified using flanking primers and 15 PCR cycles. The resulting PCR product was purified and sequenced on an Illumina MiSeq with 150 bp paired-end reads. The resulting sequencing files were quality-cleaned using TrimmGalore with a minimum Phred score of 20 and adapter trimming. Quality control was assessed with FastQC and MultiQC. Sequences were mapped with STAR to the initial IRES library sequences, and sequences with less than 88% overall mapping coverage and fewer than 2 raw counts were excluded from downstream analyses. We performed exploratory data analysis to assess the distribution of reads and identify potential biases in representation across samples (Gini coefficient [0.98], Spearman correlation [$\rho \approx 0.58$], and top-10% IRES contribution), finding that a small subset of RNAs dominated the total read counts. Based on these observations, normalization was performed to improve comparability between conditions. Read counts were normalized using the trimmed mean of M-values (TMM) method in EdgeR to adjust for differences in sequencing depth and RNA composition. Normalized counts were then used to calculate $\log_2$ fold changes ($\log_2$FC) between GFP and mRuby3 versus GFP-only populations to identify IRESs enriched under basal and thapsigargin-treated conditions. All sequencing data are deposited in Jan, Eric; Chapagain, Subash (2025), "Discovery of functional factorless internal ribosome entry site-like structures through virome mining", Mendeley Data, V1, https://doi.org/10.17632/3rwdwcrfhw.1.

## Phylogenetic analysis

Phylogenetic relationships among dicistrovirus genomes were analyzed using maximum likelihood methods implemented in IQ-TREE (version 2.0.7). RdRP sequences from genomes >6 kb were aligned using MUSCLE v3.8 with the parameters -maxiters 1 -diags1 -sv. The optimal substitution model used was Blosum62+I+G4 as determined by the ModelFinder in IQ-TREE. A phylogenetic tree was constructed with 1,000 ultrafast bootstrap replicates to provide branch support. RdRp sequences from the Hepatitis A virus (AWK22878.1), Heterosigma akashiwo RNA virus (NP_944776.1), and Nilaparvata lugens honeydew virus-1 (YP_009505599.1) were used as outgroups based on published RdRp-bases phylogenetic trees.

## Dicistrovirus genomes selection and filtering

Viral genomes were selected from multiple sources, including the Serratus database, NCBI Viral genomes, and published metatranscriptomes. Dicistrovirus-like genomes were identified by screening RNA-dependent RNA polymerase

(RdRP) sequences from known ICTV-classified dicistroviruses using Palmscan. Initially, 64,245 dicistrovirus-like genomes were identified. To minimize redundancy, genomes were clustered at 98% nucleotide identity, and sequences shorter than 2,000 bp were excluded. This filtering resulted in a final list of 9,151 genomes, which were then analyzed for IRES identification.

### IRES search

RNA secondary structure models were generated to search for Type IRESs in a set of dicistrovirus genomes using INFERNAL v1.1.3. The INFERNAL search was performed using default parameters. 6a subtype IRES covariance models were generated from IGR IRESs of Drosophila C virus (DCV - AF014388), Homalodisca coagulata virus 1 (HoCV1 - DQ288865), aphid lethal paralysis virus (ALPV - AF536531), Rhopalosiphum padi virus (RhPV - AF022937), Triatoma virus (TrV - AF178440), Cricket paralysis virus (CrPV - AF218039), Black queen cell virus (BQCV - AF183905), Himetobi P virus (HiPV - AB017037), and Plautia stali intestine virus (PSIV – AB006531). 6b subtype IRES covariance models were generated from IRESs of the Acute bee paralysis virus (ABPV – AF150629), Israeli acute paralysis virus (IAPV – EF219380), Kashmir bee virus (KBV AY275710), Taura syndrome virus (TSV – AF277675), and Solenopsis invicta virus 1 (SINV1 – AY634314). 6c subtype IRES covariance models were generated from IRESs the Kuiper virus (KX657785.1), Changjiang picorna-like virus 14 (NC_032773.1), *Halastavi árva* RNA virus (NC_016418.1), and Shahe arthropod virus 1 strains SHWC01c3692 (KX883988). 6d subtype IRES covariance models were generated from IRESs of the Antarctic picorna-like virus 1 (APLV1 - KM259869.1), Beihai picorna-like virus 78 (KX883307.1), Changjiang picorna-like virus 9 (KX884541.1), Gingko biloba dicistrovirus strain pt112-dic-11 (MN729613.1), nedicistrovirus (NediV - JQ898341.1), Sanxia picorna-like virus 12 (KX883723.1), s64-k141_ 2464283 (MZ678982.1), Picornavirales sp. isolate R35-k141_316374(MZ678988.1), and *Pallasea cancelloides* TSA sequence (GEQX01016691.1). 6e subtype IRES covariance models were generated from IRESs of the Caledonia beadlet anemone dicistro-like virus 1 (MF189971.1), Beihai picorna-like virus 85 (KX883346), bivalve RNA virus G5 (NC_032115), and *Proasellus solanasi* 2 (HAFJ01050538.1). 6f subtype IRES covariance models were generated from the Changjiang picorna-like virus 9 (KX884560.1), Changjiang picorna-like virus 6 (KX884548.1), Wenzhou picorna-like virus 31 (KX884341.1), and *Lactuca sativa* dicistroviridae strain pt151 (MN722414.1). The identified IRESs were manually curated to remove alignment gaps, and the sequences were classified into subtypes based on scores and structural similarity. All datasets are shown in S1–S5 Data.

### Supporting information

**S1 Data. List of genomes used for IGR IRES structure search.**
(XLSX)

**S2 Data. Predicted IRES structures used for Fig 2 - Consensus IRES sequences.**
(XLSX)

**S3 Data. IRES sequences used for lentivirus library for Fig 3A and 3B.**
(XLSX)

**S4 Data. Predicted IRES structures selected for functional analysis (Figs 3, 4, 6, 7 and 8).**
(XLSX)

**S5 Data. Genomic architectures of dicistroviruses containiing predicted IRES structures (Figs 5 and S5).**
(XLSX)

**S6 Data. SAXS analysis of non-DV-1 RNA.**
(PDF)

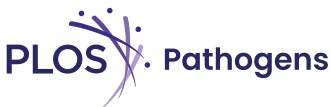

**S1 Fig. Analysis of lentiviral IRES screen.**
(TIF)

**S2 Fig. Model of Type 6 IRES structures identified by INFERNAL.**
(PDF)

**S3 Fig. Distribution of predicted Type 6 IRESs in typical and atypical dicistrovirus genomes.**
(TIF)

**S4 Fig. Translational activities of predicted Type 6 IRES structures from non-dicistroviruses.**
(TIF)

**S5 Fig. SAXS and SHAPE analysis of predicted non-DV1 IRES structure.** Detailed description of SAXS analysis.
(TIF)

## Acknowledgments

We thank the Jan lab members for feedback and comments. We acknowledge UBC Sequencing and Bioinformatics Consortium for amplicon sequencing and analysis. We would additionally like to thank the B21 beamline scientists at Diamond Light Source for their continual support.

## Author contributions

**Conceptualization:** Nicolas Salcedo-Porras, Nozomu Yachie, Eric Jan.

**Data curation:** Nicolas Salcedo-Porras, Amir Abdolahzadeh, Stephane Flibotte, Artem Babaian.

**Formal analysis:** Subash Chapagain, Nicolas Salcedo-Porras, Amir Abdolahzadeh, Yaohua Zhang, Higor Sette Pereira, Stephane Flibotte, Kevin Low, Christina Young, Yuhang Wu, Shao Wang, Soh Ishiguro, Eric Jan.

**Funding acquisition:** Trushar Patel, Eric Jan.

**Investigation:** Subash Chapagain, Nicolas Salcedo-Porras, Amir Abdolahzadeh, Yaohua Zhang, Higor Sette Pereira, Stephane Flibotte, Kevin Low, Christina Young, Yuhang Wu, Shao Wang, Soh Ishiguro, Nozomu Yachie, Trushar Patel, Eric Jan.

**Methodology:** Subash Chapagain, Nicolas Salcedo-Porras, Amir Abdolahzadeh, Yaohua Zhang, Higor Sette Pereira, Stephane Flibotte, Kevin Low, Christina Young, Yuhang Wu, Shao Wang, Soh Ishiguro, Eric Jan.

**Project administration:** Nozomu Yachie, Trushar Patel, Artem Babaian, Eric Jan.

**Resources:** Nicolas Salcedo-Porras, Amir Abdolahzadeh, Stephane Flibotte, Nozomu Yachie, Trushar Patel, Artem Babaian, Eric Jan.

**Software:** Nicolas Salcedo-Porras, Stephane Flibotte, Artem Babaian.

**Supervision:** Nozomu Yachie, Trushar Patel, Artem Babaian, Eric Jan.

**Validation:** Subash Chapagain, Nicolas Salcedo-Porras, Amir Abdolahzadeh, Yaohua Zhang, Higor Sette Pereira, Stephane Flibotte, Kevin Low, Christina Young, Yuhang Wu, Shao Wang, Soh Ishiguro, Eric Jan.

**Visualization:** Amir Abdolahzadeh, Yaohua Zhang, Higor Sette Pereira, Stephane Flibotte, Kevin Low, Eric Jan.

**Writing – original draft:** Stephane Flibotte, Eric Jan.

**Writing – review & editing:** Subash Chapagain, Amir Abdolahzadeh, Higor Sette Pereira, Stephane Flibotte, Christina Young, Nozomu Yachie, Eric Jan.



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
