## [Decision Letter · Decision Letter 0]

Discovery of functional factorless internal ribosome entry sites through virome mining

PLOS Pathogens

Dear Eric,

Thank you for submitting your manuscript to PLOS Pathogens. After careful consideration, we feel that it has merit but does not fully meet PLOS Pathogens's publication criteria as it currently stands. Therefore, we invite you to submit a revised version of the manuscript that addresses the points raised during the review process.

Please submit your revised manuscript within 60 days Apr 03 2025 11:59PM. If you will need more time than this to complete your revisions, please reply to this message or contact the journal office at plospathogens@plos.org. Please include the following items when submitting your revised manuscript:

We look forward to receiving your revised manuscript.

Kind regards,

Peter Sarnow

Academic Editor

PLOS Pathogens

Alexander Gorbalenya

Section Editor

PLOS Pathogens

Editor-in-Chief

PLOS Pathogens

orcid.org/0000-0003-2946-9497

Michael Malim

Editor-in-Chief

PLOS Pathogens

orcid.org/0000-0002-7699-2064

**Additional Editor Comments :**

As you can see, all three reviewers noted that you're studying an important problem. However, they all made several comments (especially reviewer #3) that will improve the paper. Kindly address them carefully in a revision.

**Journal Requirements:**

1) We noticed that you used the phrase 'data not shown' in the manuscript. We do not allow these references, as the PLOS data access policy requires that all data be either published with the manuscript or made available in a publicly accessible database. Please amend the supplementary material to include the referenced data or remove the references.

4) Please amend your detailed Financial Disclosure statement. This is published with the article. It must therefore be completed in full sentences and contain the exact wording you wish to be published.

1) Please clarify all sources of financial support for your study. List the grants, grant numbers, and organizations that funded your study, including funding received from your institution. Please note that suppliers of material support, including research materials, should be recognized in the Acknowledgements section rather than in the Financial Disclosure

2) State the initials, alongside each funding source, of each author to receive each grant. For example: "This work was supported by the National Institutes of Health (####### to AM; ###### to CJ) and the National Science Foundation (###### to AM)."

3) State what role the funders took in the study. If the funders had no role in your study, please state: "The funders had no role in study design, data collection and analysis, decision to publish, or preparation of the manuscript."

5) Your current Financial Disclosure states, "The author(s) received no specific funding for this work.".

However, your funding information on the submission form indicates receiving funds. Please ensure that the funders and grant numbers match between the Financial Disclosure field and the Funding Information tab in your submission form. Note that the funders must be provided in the same order in both places as well.. 

Please indicate by return email the full and correct funding information for your study and confirm the order in which funding contributions should appear. Please be sure to indicate whether the funders played any role in the study design, data collection and analysis, decision to publish, or preparation of the manuscript.

**Reviewers' Comments:**

Reviewer's Responses to Questions

**Part I - Summary**

Reviewer #1: The manuscript by Chapagain et al., reports the computational search of sequences in viral genomes potentially acting as internal ribosome entry site elements (IRES) resembling the factorless IRES of Dicistrovirus genomes, and classify them in subtypes according to predicted structure. The manuscript shows the validation of some of the identified elements in vivo and in vitro. The ribosome binding ability of some of these RNA sequences is also shown. Interestingly, the genomic location of these novel IRESs shows various atypical settings, including IRESs embedded within an ORF. Finally, some of the IREs-like structural elements were found in Tombusvirus and Narnavirus genomes, previously unknown to initiate translation via IRES elements. Overall, the manuscript provides significant amount of novel information, taking advantage of computational methods to search for novel RNA structural elements. However, the manuscript is far too long. It can be significantly summarized to avoid extra interpretations, and also to avoid repetitions between results and discussion.

Reviewer #2: The positive-sense RNA genomes of dicistroviruses possess the smallest and most compact IRESs between the ORF1, which encodes an RNA-dependent RNA polymerase polyprotein, and ORF2, which encodes capsid proteins (IGR or type 6 IRESs). A salient feature of IGR IRESs is their ability to initiate translation without the assistance of initiation factors, an initiation codon, and initiator tRNA, representing the simplest mechanism of initiating eukaryotic protein synthesis.

Chapagain et al. used a bioinformatics approach to identify type 6 IRES-like structures in viral RNA genomes. This analysis revealed 4,704 dicistrovirus-like genomes, possessing 3257 type 6 IRES-like structures (mostly in genomes >6 kb). Among them were previously described subtype 6a and 6b IRESs, which validates the undertaken approach. IRES consensus models were constructed for the types 6a, 6b, 6d, 6e, and 6f IRESs and used to examine functional elements within these IRESs. Most of the predicted 6a and 6b IRES structures were functional in cells and cell-free extracts, with the integrity of PKI base pairing being critical for activity. All tested IRESs bound directly to 80S ribosomes as demonstrated using filter binding assays. Interestingly, they also identified several atypical architectures of dicistrovirus genomes and localizations of IRESs. Of particular interest is a subset of IRESs where the upstream ORF1 stop codon is located within or downstream of the IRES. In this case, the translation of ORF1 reduced IRES-directed translation, apparently through the disruption of IRES structure by translating ribosomes. Finally, they identified IRESs within non-dicistrovirus genomes and showed their ability to bind ribosomes directly. It is puzzling that 80S complexes with some IRES-like structures exhibited elongation incompetence. I hope they will address this conundrum in the follow-up studies.

Overall, this is an interesting and important study that demonstrates the utility of the undertaken bioinformatics approach for the identification of novel type 6 IRES-like structures. Importantly, these IRESs were found in typical and atypical locations and even within viral open reading frames. The results are convincing, as being supported by numerous controls. The paper will interest virologists and molecular biologists investigating the mechanism of protein synthesis.

Reviewer #3: In this study, Chapagain, Salcedo-Porras, Abdolahzadeh, et al. developed a bioinformatics pipeline based on the Infernal software, which uses RNA-specific sequence and structural features to identify Type 6 IRES structures. Furthermore, they selected and validated the function of some of these predicted IRES-like structures using bicistronic RNAs in vitro and cells. The authors use this integrated pipeline to identify RNAs with IRES-like structures and report novel IRESs. Interestingly, the study shows that not all IRES-like RNA structures enable translation initiation. Also, results suggest that not all identified IRESs are factor-free, as their function is deeply affected by the environment (in RRL, Sf21 lysates, or cells).

Data are very interesting, and the developed bioinformatic pipeline is attractive to those studying viral IRESs whose function relies only on RNA structure. However, data interpretation in some text sections is highly speculative and sometimes could be misleading to a general reader, as expected from PLoS Pathogens.

**Part II – Major Issues: Key Experiments Required for Acceptance**

Reviewer #1: The manuscript shows that many dicistrovirus genomes (<6kb in length) do not contain IRES-like structures in the intergenic region. What type of RNA sequences/structures were found in these genomes?

Figure 3. I am surprised that the authors did not check the potential cryptic promoter activity of the sequences tested in figure 3A,B. The overall similar read counts of the thapsigargin treated samples relative to untreated conditions is striking (Fig 3C). It seems that this compound increases basal translation activity in a rather unspecific manner. Additionally, there are large differences in the activity measured in RRL compared to Sf21 cells (Fig 3D), some of them being negative in one of the systems. Perhaps, all these elements should be classified according to their activity in vitro or in vivo.

Figure 4: Curiously, type 6a-11 used to generate mutations within PKII and PKIII to analyze Ribosome binding was not assayed in Fig 3D. Authors should comment why this particular IRES is used.

The main text can be significantly summarized to eliminate overinterpretations, and also to circumvent repetitions between results and discussion.

Reviewer #2: There are no major issues.

Reviewer #3: Line 190-on: The authors developed a covariance model using the six known Type 6 IRES subtypes and used INFERNAL to scan the dicistrovirus genome database. As a result, a number of IRES-like RNA structures are identified. Among the hits, several known RNA structures with IRES activity are recognized, validating the search strategy. The authors also identified RNA structures in the dicistrovirus genome database that have not previously shown IRES activity. Authors should remember that they have identified RNA structures but not IRESs, as IRESs are defined by their function. The text should be modified accordingly. The link between IRES-like RNA structure discovery and IRES activity has not yet been made for all identified IRES-like RNA structures in the manuscript. Based on the findings, generalization seems inadequate.

Lines 223-226, data do not warrant the author's conclusions "….thus suggesting that more than half of the dicistroviruses may utilize an alternative structure and/or translation mechanism to drive downstream ORF expression". The comment is highly speculative. The authors are analyzing a sequence database. Do they have data demonstrating that all analyzed sequences are functional regarding ORF expression? If so, please share or cite. It is correct to associate findings with a translation mechanism only if proteins are synthesized. Furthermore, as stated in the text, the lack of RNA elements within certain sequences …" could be attributed to incomplete or truncated genomes".

Figure 1C "IRES count" in the graph axis is misleading; at this point in the manuscript, IRES activity has not been demonstrated for the identified hits.

Figure 1E, without IRESs, with IRESs, should be changed; what is being classified are RNA structures that share structural features with a dicistrovirus IRES. No IRES function has been demonstrated for these RNA sequences.

Line 227-250. To gain insights into the evolution of Type 6 IRESs, the authors constructed an RdRP-based phylogenetic tree of the dicistrovirus genomes (>6 kb) and plotted the IRES-like RNA structures in this tree. Definitive conclusions can only be drawn from RNA structures with demonstrated IRES activity. Using the newly identified IRES-like RNA structures is highly speculative as no proof has been provided that they are indeed translationally functional.

Line 253-255. The phrase "To examine functional elements within the Type 6 IRESs, we aligned unique high-scoring IRES subtypes through multiple sequence alignments to construct consensus models" is unclear. The authors have analyzed sequences and identified RNA structures. Functional elements in the context of the IRES-like RNA structure can not be defined unless the IRES is proven to be functional.

Line 289-291. Conclusion… "In general, these analyses highlighted the length of the base-paired regions and key nucleotide identity conservation that are likely important for IRES function", is only valid for sequences known to assemble in a functional IRES. Adding "candidate IRESs" to the analysis is incorrect and does not warrant the conclusion.

292-303. The authors analyze predicted Type 6 IRESs and observe predicted initiation codons. Yet conclude, "These analyses indicated a strong selection of start codons for IGR IRES-mediated viral protein synthesis". Are all the predicted Type 6 IRESs functional? In fact, later in the text, they show that few IRESs are active. From all, only 209 IRESs were identified.

Flow cytometry analysis identified 4 populations of cells: GFP low/mRuby3 low, GFP high//mRuby3 low, GFP low/mRuby3 high, and GFP high/mRuby3 high. Thapsigargin treatment increased two populations: GFP low/mRuby3 high and GFP high/mRuby3 high; only one is mentioned in the text. If what is interesting for the authors in IRES activity, why were the GFP low/mRuby3 high cells ignored? Doesn't ER stress suppress cap-dependent translation initiation? If so, why does the population of GFP high cells increase in the presence of Thapsigargin? Why is gating not equivalent for Thapsigargin-treated and non-treated cells?

The description of Figure 3 D in the text does not seem accurate. From the tested putative type 6a IRESs (1 -10), 6, 7,8, 9, and 10 showed no IRES activity in RRL. From the putative type 6b IRESs (1-9), 7, 8, 9 showed no IRES activity in RRL., Not all putative IRESs were tested in Sf21 lysates; why? From the tested putative IRESs, in Sf21 lysates 6a (3,4,5,6) and 6b-4 had no activity.

Interestingly, authors identify novel IRESs using a bioinformatic pipeline based on the structure of a known and well-characterized factorless IRES. However, this identification strategy does not directly imply that all discovered IRESs are factorless, as suggested in the manuscript's title and throughout the text. Data indicate that the function of most of the identified IRESs depends on the overall environment (RRL vs. Sf21 or Thapsigargin treatment or not), which most likely suggests that they depend on factors for their function.

Data shown in Figure 3D and Figure 4A are inconsistent, as the Fluc/Rluc ratio for the wt IRESs are not comparable. For example, in Figure 3D 6a1 wt, RTA is about 2,5 relative to the CrPV wt set to 1, while RTA for 6b1 wt has about 4. In Figure 3D, the RTA for 6a3 wt is equivalent to that of 6a2 wt and higher than the CrPV wt control RTA. Both experiments used the same bicistronic RNA and were conducted in RRL. Normalization in Figure 3D and Figure 4A was done the same way. So, data should be equivalent for the wt. For experiments shown in Figures 3D and 4A, raw Rluc and Fluc data should be shown to better understand the conclusions.

The interpretation of Figure 4A seems incorrect. The text says …." As predicted, disruption of the predicted PKI base-pairing abolished IRES translation whereas compensatory mutations that restored PKI base-pairing partially rescued IRES activity". Data clearly shows that mutations in PKI abolish IRES activity. What is unclear in the partial rescue associated with the compensatory mutations?

In Figure 4C the authors introduce the putative IRES 6a-11. Information regarding this putative IRES was not included in previous figures 3D or 4A. First, they show 80S binging, and then they show IRES activity. The order does not make much sense with the rest of the text. Kd of 80S binding for 6a-11 should also be included. They should also show the activity of 6a-11 in RRL.

In the section "Novel dicistrovirus genomic architectures" agin the authors should consider that what they are identifying are IRES like RNA structures and not functional IRESs. Their method undoublty works to narrow days structures with putative IRES activity yet not all identified structureds do have IRES activity. As presented this section seems highly speculative as no functional assay for all identified IRES like structures is presented.

Conclusions regarding Figure 6 are unclear. The authors suggest that ribosome translating through the overlapping IRES unwinds the IRES structure, reducing IRES activity. Even though this is possible, the data do not warrant the conclusion. In these experiments, populations of RNA are evaluated, so what happens at a single molecule level remains unknown. What if a single translating RNA can load ribosomes either using a cap- or an IRES mechanism? But cannot use both mechanisms simultaneously. Observations could reflect the competition between cap- and IRES-dependent translation. If so, data show that the IRES is efficient in its recruitment process. We can also see that the cap is functional. However, when the HP abolishes cap-dependent translation initiation, the IRES-mediated mechanism loses its competitor, and IRES activity takes over. Positioning the stop codon before or after the IRES would impact the time taken for the mRNA to reinitiate a new round of translation. The earlier it is released for the translational apparatus, the more it can be used and the more protein it will synthesize. The latter would be reflected in an apparent increase in the RTA. With the provided data, how can the authors discard this latter possibility?

Data in Figure 6D show that when cells are transfected with the bicistronic RNA containing the stop codon after RLuc (RS), it results in higher IRES-mediated translation than the reporter RNA containing the 6a-2 IRES in its natural context (WT). Authors should be cautious when interpreting this data as they have not included the controls to conclude that translation, though the overlapping IRES impacts IRES-mediated translation. What if, in cells, their system generates a monocistronic RNA that conserves IRES activity encoding for Fluc? In this case, laking a stop codon would increase Fluc activity.

Regarding Figure 6D, why does cap-dependent translation increase in CrPV-infected cells?

Lines 475-476… "An unexpected finding from the bioinformatics search was the discovery of embedded Type 6 IRESs within the viral monocistronic open frame"…putative Type 6 IRESs.

Figure 7A-D, Why is cap-dependent translation so low? In figure 6, the authors suggest that ribosome translating through the overlapping IRES unwinds the IRES structure, reducing IRES activity. This model is not consistent with data obtained for Contignous IRESs (6a3, 6a9, 6a11 to 6a18, 6b8, and type 6b11 and 12). Adding a stop codon after RLuc does not increase Fluc from 6a11(Fig.7D).

In Figure 7D, assuming the equivalent amounts of RNA are translated, why do mutations in the 6a-11 IRES impact cap-dependent translation? Compare [35S] with eth wt.

Lines 491-493. "…..for the majority of contiguous IRES containing reporter RNAs in vitro, suggesting that both scanning-dependent and IRES-mediated translation may be occurring within the same mRNA." The conclusion is unclear, as experiments are conducted with a population of mRNAs.

Lines 494-496. "Moreover, a reporter RNA with contiguous IRES 6b-8, resulted in expression of a truncated RLuc (36 kDa protein) which is suggestive that ribosomes may be dropping off upon reaching the IRES structure". These comments are inconsistent with the shown data, as no truncated form of RLuc is generated from contiguous IRES 6b-8.

Regarding Figure 7F lane 1, the laddering of proteins is easy to see. Yet, the laddering of protein multiples of ~20 kDa is not evident compared to the MW ladder. Lane 2 NLuc is barely seen. Why? Can this be improved? As presented data are not consistent with what is shown in cells Fig.7G.

Data using the circular RNA in cells is difficult to understand and is not consistent with the data obtained in RRL. In cells, the mPKI-NLuc.circ is functional, working as efficiently as the WT-NLuc-circ. This suggests that an NLuc.circ RNA harboring an irrelevant RNA sequence as a negative control of activity is missing. The WT-NLuc stop-circ, which is expected to work efficiently (based on its activity in RRL), does not. In contrast to what is reported from RRL in cells, the WT-P2A-NLuc-circ is the most efficient.

Data in Figure 8 do not support the author's conclusions…. "these data revealed functional IGR-IRES like RNA elements in viral genomes other than dicistroviruses and that IRES-driven mechanism may be more widespread in virosphere". The identified IRES-like RNA structures bind the 80S but do not mediate translation initiation. So, as shown in Figure 4 (6b-9) 80S assembly does not directly imply translation initiation. So, by definition, are these IRESs? What if these so-called IRESs act as 80S (or 40S) sponges in cells, reducing the 80S availability as a viral mechanism of competing with the cellular mRNAs for the components of the translational machinery?

Discussion section. Authors should consider all previous comments and criticisms (see above). Also, they should consider that what has been identified are IRES-Like RNA structures, not IRESs. Please refer to IRESs only when a function in translation has been demonstrated. They should also state if they will consider RNA elements that bind the 80S but do not mediate translation initiation as IRESs. The authors should discuss why the 80S binding (Figures 4C, 4D, 8) is not directly linked to translation initiation or IRES activity (Figure 3D). Consider data obtained with RNAs 6a1 and 6b-9 indicates.

Lines 717-718 indicate: "to our knowledge, this work is the first to demonstrate a functional embedded IRES within a viral ORF". IRESs within ORFs of viral mRNAs other than dicistroviridae have been previously described (Nucleic Acids Res. 2017;45(12):7382-7400. doi: 10.1093/nar/gkx303; Nucleic Acids Res. 2010;38(4):1367-81. doi: 10.1093/nar/gkp1109; Nat Struct Mol Biol. 2005;12(11):1001-7. doi: 10.1038/nsmb1011; J Virol. 2001;75(1):181-91. doi: 10.1128/JVI.75.1.181-191.2001).

**Part III – Minor Issues: Editorial and Data Presentation Modifications**

Reviewer #1: The authors mention twice that this study “provides a framework to map the origin of these factorless IRES mechanisms “. In my opinion, authors have identified a diversity of type 6 IRESs, but the origin of these elements is not investigated in this manuscript.

Reference # 4 by Pelletier et al., is not the correct article.

Reviewer #2: 1. Page 4, line 73 should read: “…of the family Flaviviridae …”

2. Page 11, line 224 should read: “…intergenic region were <6 kb in length…”

3. Fig.4A. The bottom part is an analysis of mRNA stability. This should be mentioned and described in the figure legend. The same critique applies to Figs.6C and 7C.

4. Page 17, line 369 should read: “(Fig.3D).”

5. Page 17, line 378 should read: “…5’ cap bicistronic RNA (Fig. 4E).”

6. Fig.6D, bottom. The difference between WT and RS 6a-2 IRES activities in CrPV-infected S2 cells is small. Is this difference significant?

7. Fig.8E legend. It is unclear whether the non-DV1 WT IRES corresponds to #1 non-dicistrovirus-like IRES from panel B, which does not support in vitro translation.

8. Page 33, line 709 should read: “…affect IRES-driven translation.”

Reviewer #3: Lines 67-69 reads "By contrast, some positive-strand RNA viruses use structured RNAs known as internal ribosome entry sites (IRESs) promote 5′ cap-independent translation initiation", should be corrected, "By contrast, some positive-strand RNA viruses use structured RNAs known as internal ribosome entry sites (IRESs) to promote 5′ cap-independent translation initiation."

Line 76-77 reads: "……protein synthesis that can be are classified based on sequence and structural homology…."."….can be are"….should be corrected.

Line 378 reads.. "….IRES activity in insect S2 cells by transfection of 5′ cap bicistronic E)". The sentence should be completed.

Line 438 reads…" The Type 6a-2 IRES is functional (Fig. 3C)….” Should read…”The Type 6a-2 IRES is functional (Fig. 3D)”.

PLOS authors have the option to publish the peer review history of their article (what does this mean? ). If published, this will include your full peer review and any attached files.

**Do you want your identity to be public for this peer review?** For information about this choice, including consent withdrawal, please see our Privacy Policy .

Reviewer #1: No

Reviewer #2: No

Reviewer #3: No

**Figure resubmission:**

**Reproducibility:**



---

## [Decision Letter · Decision Letter 1]

PPATHOGENS-D-24-02857R1

Discovery of functional factorless internal ribosome entry site-like structures through virome mining

PLOS Pathogens

Dear Eric,

As you can see, all reviewer were satisfied with your editions. Can you kindly address the comments made by reviewer #2 about Figure 6 and re-submit?

Please submit your revised manuscript within 30 days Jul 25 2025 11:59PM. If you will need more time than this to complete your revisions, please reply to this message or contact the journal office at plospathogens@plos.org. Please include the following items when submitting your revised manuscript:

We look forward to receiving your revised manuscript.

Kind regards,

Peter Sarnow

Academic Editor

PLOS Pathogens

Alexander Gorbalenya

Section Editor

PLOS Pathogens

Sumita Bhaduri-McIntosh

Editor-in-Chief

PLOS Pathogens

orcid.org/0000-0003-2946-9497

Michael Malim

Editor-in-Chief

PLOS Pathogens

orcid.org/0000-0002-7699-2064

**Reviewers' Comments:**

Reviewer's Responses to Questions

**Part I - Summary**

Reviewer #1: The response is satisfactory

Reviewer #2: The revised manuscript by Chapagain et al. has been improved. Considering that IRESs are defined by function and not by structure (as pointed out by reviewer #3), they modified the text by substituting IRESs with IRES-like structures where functional evidence is lacking. In addition, they improved some figures. However, there are still some issues. In particular, Fig. 6 does not make a convincing case.

Reviewer #3: In this study, Chapagain, Salcedo-Porras, Abdolahzadeh, et al. developed a bioinformatics pipeline based on the Infernal software, which uses RNA-specific sequence and structural features to identify Type 6 IRES structures. Furthermore, they selected and validated the function of some of these predicted IRES-like structures using bicistronic RNAs in vitro and cells. The authors use this integrated pipeline to identify RNAs with IRES-like structures and report novel IRESs. Interestingly, the study shows that not all IRES-like RNA structures enable translation initiation. Also, results suggest that not all identified IRESs are factor-free, as their function is deeply affected by the environment (in RRL, Sf21 lysates, or cells).

**Part II – Major Issues: Key Experiments Required for Acceptance**

Reviewer #1: -

Reviewer #2: Fig. 6. Their data do not support the idea that ribosomes while moving through overlapping IRES relax the secondary structure and reduce IRES activity. As shown in Fig.6D, the difference between the numbers of Fluc units produced in WT and RS mRNA-transfected S2 cells is small and insignificant. Furthermore, by eye, the WT, RS, and HP mRNAs yield similar amounts of the Fluc protein upon translation in RRL (Fig.6C). A higher Fluc/Rluc ratio for HP mRNA translation does not mean much as this most likely reflects the weakening of the Rluc band. They should have presented raw data for the Fluc band intensity. Finally, they should not ignore the possibility of competition between cap- and IRES-dependent translations (as mentioned by reviewer #3). So, they can only conclude that the 6a-2 IRES structure in the WT mRNA retains activity despite its possible unwinding by translating ribosomes or competition from cap-dependent translation.

Reviewer #3: Major concerns raised in the first round of revision were appropriately addressed in the revised version of the manuscript.

**Part III – Minor Issues: Editorial and Data Presentation Modifications**

Reviewer #1: -

Reviewer #2: Line 70 should read: “…been reported in retroviruses HIV-1 and HIV-2…”

Line 220-221. I disagree with their response. The correct statement should be: “Similarly, the majority of dicistrovirus genomes that do not contain a Type 6 IRES-like structure within the intergenic region were <6 kb in length”.

Line 1439 should read: “mPKI-III denotes mutations…”

Reviewer #3: Minor concerns raised in the first round of revision were appropriately addressed in the revised version of the manuscript.

PLOS authors have the option to publish the peer review history of their article (what does this mean? ). If published, this will include your full peer review and any attached files.

**Do you want your identity to be public for this peer review?** For information about this choice, including consent withdrawal, please see our Privacy Policy .

Reviewer #1: No

Reviewer #2: No

Reviewer #3: No

**Figure resubmission:**
---

## [Editor Report · Decision Letter 2]

Dear Eric,

We are pleased to inform you that your manuscript 'Discovery of functional factorless internal ribosome entry site-like structures through virome mining' has been provisionally accepted for publication in PLOS Pathogens.

Best regards,

Peter Sarnow

Academic Editor

PLOS Pathogens

Alexander Gorbalenya

Section Editor

PLOS Pathogens

Sumita Bhaduri-McIntosh

Editor-in-Chief

PLOS Pathogens

orcid.org/0000-0003-2946-9497

Michael Malim

Editor-in-Chief

PLOS Pathogens

orcid.org/0000-0002-7699-2064
---

## [Editor Report · Acceptance letter]

Dear Dr. Jan,

We are delighted to inform you that your manuscript, "Discovery of functional factorless internal ribosome entry site-like structures through virome mining," has been formally accepted for publication in PLOS Pathogens.

Best regards,

Sumita Bhaduri-McIntosh

Editor-in-Chief

PLOS Pathogens

orcid.org/0000-0003-2946-9497

Michael Malim

Editor-in-Chief

PLOS Pathogens

orcid.org/0000-0002-7699-2064